# UNCOVERING CHALLENGES OF SOLVING THE CONTINUOUS GROMOV-WASSERSTEIN PROBLEM

## ABSTRACT

Recently, the Gromov-Wasserstein Optimal Transport (GWOT) problem has attracted the special attention of the ML community. In this problem, given two distributions supported on two (possibly different) spaces, one has to find the most isometric map between them. In the discrete variant of GWOT, the task is to learn an assignment between given discrete sets of points. In the more advanced continuous formulation, one aims at recovering a parametric mapping between unknown continuous distributions based on i.i.d. samples derived from them. The clear geometrical intuition behind the GWOT makes it a natural choice for several practical use cases, giving rise to a number of proposed solvers. Some of them claim to solve the continuous version of the problem. At the same time, GWOT is notoriously hard, both theoretically and numerically. Moreover, all existing continuous GWOT solvers still heavily rely on discrete techniques. Natural questions arise: to what extent do existing methods unravel the GWOT problem, what difficulties do they encounter, and under which conditions are they successful? Our benchmark paper is an attempt to answer these questions. We specifically focus on the continuous GWOT as the most interesting and debatable setup. We crash-test existing continuous GWOT approaches on different scenarios, carefully record and analyze the obtained results, and identify issues. Our findings experimentally testify that the scientific community is still missing a reliable continuous GWOT solver, which necessitates further research efforts. As the first step in this direction, we propose a new continuous GWOT method which does not rely on discrete techniques and partially solves some of the problems of the competitors.

## 1 INTRODUCTION

Optimal Transport (OT) is a powerful framework that is widely used in machine learning (Montesuma et al., 2023). A popular application of OT is the domain adaptation of various modalities, including images (Courty et al., 2016; Luo et al., 2018; Redko et al., 2019), music transcription (Flamary et al., 2016), color transfer (Frigo et al., 2015), alignment of embedding spaces (Chen et al., 2020; Aboagye et al., 2022). Other applications include generative models (Salimans et al., 2018; Arjovsky et al., 2017), unpaired image-to-image transfer (Korotin et al., 2023b; Rout et al., 2022), etc.

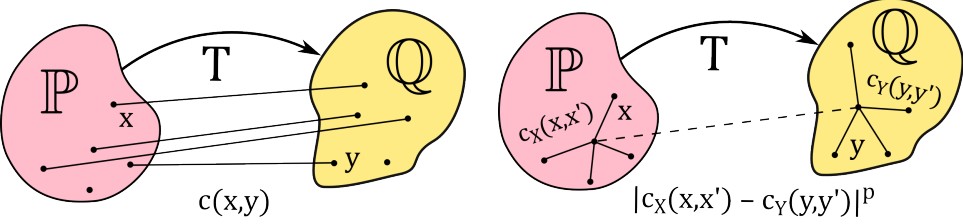

(a) Monge's OT between distributions $\mathbb{P}$ and $\mathbb{Q}$ with *inter-domain* cost function $c(x, y)$.

(b) Monge's GW between distributions $\mathbb{P}$ and $\mathbb{Q}$ with *intra-domain* costs $c_{\mathcal{X}}(x, x')$ and $c_{\mathcal{Y}}(y, y')$.

Figure 1: A schematic visualization of the OT problems and GW problems (Monge's form).

In the conventional OT problem (Figure 1a), one needs to find a map between two data distributions that minimizes a certain "effort" expressed in the form of an *inter-domain* transport cost function. This cost function shows how hard it is to move a point in the source space to a given point in the target space. Thus, in order for the resulting map to possess certain useful properties, one has to

incorporate them into the cost function. Unfortunately, this is not always a straightforward task, especially when the data distributions are supported in different spaces.

A popular way to address the above-mentioned issue is to consider the Gromov-Wasserstein (GW) modification (Memoli, 2007; Mémoli, 2011; Peyré et al., 2016) of the OT problem (Figure 1b). Here one assumes that both the source and target spaces are equipped with a structure, e.g., with a metric, and one aims to find a transport map that maximally preserves this structure, i.e., the most isometric map. This clear geometrical intuition behind GW makes it natural in various applications: unsupervised data alignment (Alvarez-Melis & Jaakkola, 2018; Aboagye et al., 2022), single-cell data processing (Scetbon et al., 2022; Klein et al., 2023; Sebbouh et al., 2024; Demetci et al., 2024; Ryu et al., 2024), 2D and 3D shape analysis (Beier et al., 2022; Mémoli, 2009), graph data analysis (Xu et al., 2019; 2021; Vincent-Cuaz et al., 2022; Chowdhury & Needham, 2021; Vincent-Cuaz et al., 2021; Mazelet et al., 2025) and dimensionality reduction (Assel et al., 2024).

**Discrete/continuous GW**. The GWOT problem is about learning some specific translation that operates with source and target distributions. In practice, these distributions are typically given by empirical datasets. This leads to two possible ways of paving a GWOT map. In the **discrete** scenario, the learned translation is just a point(s)-to-point(s) assignment (transport matrix). In turn, the **continuous** GW is about learning a parametric mapping between the underlining (continuous) distributions. In this case, the datasets are treated as i.i.d. samples derived from them.

While existing computational approaches for the GW problem show considerable empirical success, the problem itself is highly non-trivial from different perspectives.

- **Theory.** Finding the most isometric map between probability spaces based just on the inner properties of these spaces may be poorly defined, e.g., the desired transform may be non-unique. This happens where the source (target) space permits some non-trivial isometries that preserve the corresponding source (target) distribution. A simple yet expressive example is the Gaussian case Delon et al. (2022). Intuitively, non-uniqueness may affect the stability of a GWOT solver.

- **Computations/algorithms.** It is known that the discrete GW yields a non-convex quadratic optimization problem (Vayer, 2020), which is computationally challenging. To partially alleviate the difficulty, one typical approach is to consider entropic regularization (Peyré et al., 2016; Alvarez-Melis & Jaakkola, 2018; Scetbon et al., 2022; Wang & Goldfeld, 2023). Fortunately, the regularized problem resorts to a sequence of tractable Sinkhorn OT assignments. However, the convergence of the procedure may not hold, see (Peyré et al., 2016, Remark 3). In addition, discrete GWOT scales poorly with the number of input (source or target) samples, making some problem setups unmanageable by such solvers. While there are techniques to reduce the computational burden w.r.t. data size (Scetbon et al., 2022), they come at the cost of additional restrictions and assumptions.

- **Methodology.** Majority of existing continuous GW methods are based on discrete GW techniques and inherit all the computational challenges of the latter. Moreover, the transition from the discrete to the continuous setup may be questionable from a statistical point of view (Zhang et al., 2024).

Having said that, one naturally wonders: how do the current continuous GWOT methods manage to overcome these problems and show good practical results? What are the "bad cases" under which the aforementioned difficulties become critical and the solvers struggle? How to fight with these "bad cases"? In our paper, we shed light on these GWOT methods' ambiguities, specifically focusing on the continuous setup for the inner product and cosine distance cost functionals. **Our contributions:**

- We conduct a deep analysis of existing papers and reveal that one important characteristic that may greatly affect practical performance is the considered data setup. In fact, the majority of works primarily consider datasets with some specific statistical relations between source and target samples. Formally speaking, such setups disobey the standard i.i.d. assumption on the data and may lead to ambiguous conclusions on the solvers' capabilities.

- By following the findings from the previous point, we evaluate the performance of existing continuous inner product and cosine distance based GWOT solvers in more statistically fair and practically realistic *statistically related* data setups. Our simple yet expressive experiments witness that the level of statistical relatedness indeed highly influences the quality of the learned GWOT maps. Changing the data setup may greatly deteriorate the performance of the solvers.

- To alleviate the dependence on the mutual statistical characteristics of the source and target training data, we propose a novel continuous neural GW solver operating in the inner product setting. On

the one hand, our method is not based on discrete GW. It may be learned on arbitrarily large datasets and shows reasonably good results even on the fair *statistically unrelated* data setup. On the other hand, the method is min-max-min adversarial, which negatively impacts stability and requires plenty of data for training.

Overall, our findings reveal that existing GWOT solvers can be effective for certain applications when statistically related data is available—though such conditions may be impractical for many real-world GW tasks. However, these methods fail to learn meaningful transport plans when this statistical relation is absent. To address this limitation, we propose a solver capable of handling fully statistically unrelated setups, enabling the GWOT approach to be practically useful in more realistic and challenging scenarios.

**Notations.** Throughout the paper, $\mathbb{R}^{d_x}$ and $\mathbb{R}^{d_y}$ are the source and target data spaces, respectively. The set of Borel probability distributions on $\mathbb{R}^{d_x}$ is $\mathcal{P}(\mathbb{R}^{d_x})$. The dot product of vectors $\mathbf{x}, \mathbf{x}' \in \mathbb{R}^{d_x}$ is $\langle \mathbf{x}, \mathbf{x}' \rangle_{d_x}$. For a measurable map $T: \mathbb{R}^{d_x} \to \mathbb{R}^{d_y}$, we denote the corresponding *push-forward* operator by $T_\sharp$. For $\mathbb{P} \in \mathcal{P}(\mathbb{R}^{d_x})$ and $\mathbb{Q} \in \mathcal{P}(\mathbb{R}^{d_y})$, we denote the set of all couplings between them by $\Pi(\mathbb{P}, \mathbb{Q})$, i.e., distributions $\pi$ on $\mathbb{R}^{d_x} \times \mathbb{R}^{d_y}$ with the corresponding marginals equal to $\mathbb{P}$ and $\mathbb{Q}$.

## 2 BACKGROUND

In this section, we first explain the conventional OT setup (Villani, 2008; Santambrogio, 2015; Gozlan et al., 2017; Backhoff-Veraguas et al., 2019) and then introduce the Gromov-Wasserstein OT formulation (Mémoli, 2011; Peyré et al., 2016). Finally, we clarify our considered practical learning setup under which these problems are considered.

### 2.1 OPTIMAL TRANSPORT (OT) PROBLEM

Given two probability distributions $\mathbb{P} \in \mathcal{P}(\mathbb{R}^{d_x})$, $\mathbb{Q} \in \mathcal{P}(\mathbb{R}^{d_y})$ and a cost function $c: \mathbb{R}^{d_x} \times \mathbb{R}^{d_y} \to \mathbb{R}$, the OT problem is defined as follows:

$$\mathrm{OT}_c(\mathbb{P}, \mathbb{Q}) \stackrel{\text{def}}{=} \inf_{T_\sharp \mathbb{P} = \mathbb{Q}} \int_{\mathbb{R}^{d_x}} c(\mathbf{x}, T(\mathbf{x})) d\mathbb{P}(\mathbf{x}). \tag{1}$$

This is known as Monge's formulation of the OT problem. Intuitively, it can be understood as finding an optimal transport map $T^*: \mathbb{R}^{d_x} \to \mathbb{R}^{d_y}$ that transforms $\mathbb{P}$ to $\mathbb{Q}$ and minimizes the total transportation expenses w.r.t. cost $c$, see Figure 1a. There have been developed a lot of methods for solving OT (1) in its discrete (Cuturi, 2013; Peyré et al., 2019) and continuous (Makkuva et al., 2020; Daniels et al., 2021; Korotin et al., 2023b; Choi et al., 2023; Fan et al., 2023; Uscidda & Cuturi, 2023; Gushchin et al., 2024; Mokrov et al., 2024; Asadulaev et al., 2024) variants.

The cost function $c$ in (1) is commonly the squared Euclidean distance, making problem (1) exclusively defined for spaces of equal dimension. Handling incomparable spaces ($d_x \neq d_y$) requires a manually defined inter-domain cost, which is a nontrivial task (Wang & Zhang, 2025).

### 2.2 GROMOV-WASSERSTEIN OT (GWOT) PROBLEM

The GWOT problem is an extension of the optimal transport problem that aims to compare and transport probability distributions supported on different spaces. This problem is particularly useful when the underlying spaces do not align directly, but we still want to measure and align their intrinsic geometric structures. In what follows, we introduce the discrete and continuous variants of GWOT.

**Discrete Gromov-Wasserstein formulation.** Let $N_x$ and $N_y$ be the number of training samples in the source and target domains, respectively. Let $\mathbf{C}^x \in \mathbb{R}^{N_x \times N_x}$ and $\mathbf{C}^y \in \mathbb{R}^{N_y \times N_y}$ be the corresponding source and target intra-domain cost matrices. These matrices measure the pairwise distance or similarity between the samples for a given function, i.e., cosine similarity, Euclidean distance, inner product, etc. The discrete GWOT problem is:

$$\mathbf{T}^* \stackrel{\text{def}}{=} \underset{\mathbf{T} \in \mathcal{C}_{N_x, N_y}}{\operatorname{argmin}} \sum_{i,j,k,l} |\mathbf{C}^x_{i,k} - \mathbf{C}^y_{j,l}|^p \mathbf{T}_{i,j} \mathbf{T}_{k,l}, \tag{2}$$

where $\mathcal{C}_{N_x, N_y} \stackrel{\text{def}}{=} \{\mathbf{T} \in \mathbb{R}^{N_x \times N_y}_+ \big| \mathbf{T}^T \mathbb{1} = \frac{1}{N_y} \mathbb{1}; \mathbf{T}\mathbb{1} = \frac{1}{N_x}\mathbb{1}\}$ is the set of coupling matrices between source and target spaces; $\mathbb{1} = [1, \dots, 1]^T \in \mathbb{R}^N, N \in \{N_x, N_y\}$. The loss function $|\cdot - \cdot|^p$ in (2) is used to account for the misfit between the similarity matrices, a typical choice for the degree factor is

$p = 2$ (quadratic loss). Further details can be found in (Peyré et al., 2016; Mémoli, 2011; Chowdhury & Mémoli, 2019; Titouan et al., 2019b).

**Continuous Gromov-Wasserstein formulation.** Let $\mathbb{P} \in \mathcal{P}(\mathbb{R}^{d_x})$, $\mathbb{Q} \in \mathcal{P}(\mathbb{R}^{d_y})$ be two distributions. Let $c_{\mathcal{X}} : \mathbb{R}^{d_x} \times \mathbb{R}^{d_x} \to \mathbb{R}$ and $c_{\mathcal{Y}} : \mathbb{R}^{d_y} \times \mathbb{R}^{d_y} \to \mathbb{R}$ be two intra-domain cost functions for the source ($\mathbb{R}^{d_x}$) and target ($\mathbb{R}^{d_y}$) domains, respectively. The Monge's GWOT problem is defined as:

$$\text{GWOT}_p^p(\mathbb{P}, \mathbb{Q}) \stackrel{\text{def}}{=} \inf_{\substack{T_\sharp \mathbb{P} = \mathbb{Q} \\ \mathbb{R}^{d_x} \times \mathbb{R}^{d_x}}} \int |c_{\mathcal{X}}(\mathbf{x}, \mathbf{x}') - c_{\mathcal{Y}}(T(\mathbf{x}), T(\mathbf{x}'))|^p d\mathbb{P}(\mathbf{x}) d\mathbb{P}(\mathbf{x}'). \tag{3}$$

Theoretical results on the existence and regularity of (3) under certain cases could be found in (Dumont et al., 2024; Mémoli & Needham, 2024; 2022). An illustration of problem (3) can be found in Figure 1b. In this continuous setup, the objective is to find an optimal transport map $T^* : \mathbb{R}^{d_x} \to \mathbb{R}^{d_y}$ that allows to transform (align) the source distribution to the target distribution. While in (1) we search for a map that sends $\mathbb{P}$ to $\mathbb{Q}$ minimizing the total transport cost, (3) aims to find the most isometric map w.r.t. the costs $c_{\mathcal{X}}$ and $c_{\mathcal{Y}}$, i.e., the map that maximally preserves the pairwise intra-domain costs. The commonly studied case (Vayer, 2020; Sebbouh et al., 2024) is $p = 2$ with the Euclidean distance $c(\cdot, \cdot) = \| \cdot - \cdot \|^2$ or inner product $c(\cdot, \cdot) = \langle \cdot, \cdot \rangle$ as intra-domain cost functions. In what follows, we will use **innerGW** to denote the latter case.

## 2.3 Practical Learning Setup

In practical scenarios, the source and target distributions $\mathbb{P}$ and $\mathbb{Q}$ are typically accessible by empirical samples (datasets) $X = \{\mathbf{x}_i\}_{i=1}^{N_x} \sim \mathbb{P}$ and $Y = \{\mathbf{y}_i\}_{i=1}^{N_y} \sim \mathbb{Q}$. Under the **discrete** GWOT formulation, these samples are directly used to compute intra-domain cost matrices $\mathbf{C}^x$, $\mathbf{C}^y$. These matrices are then fed to optimization problem (2). Having been solved, problem (2) yields a coupling matrix $\mathbf{T}^*$ which defines the GWOT correspondence between $X$ and $Y$. Importantly, discrete GWOT operates with discrete empirical measures $\hat{\mathbb{P}} \stackrel{\text{def}}{=} \sum_{i=1}^{N_x} \frac{1}{N_x} \delta(\mathbf{x}_i)$, $\hat{\mathbb{Q}} \stackrel{\text{def}}{=} \sum_{i=1}^{N_y} \frac{1}{N_y} \delta(\mathbf{y}_i)$ rather than original ones.

In turn, under the **continuous** setup, the aim is to recover some parametric map $T^* : \mathbb{R}^{d_x} \to \mathbb{R}^{d_y}$ between the original source and target distributions $\mathbb{P}$ and $\mathbb{Q}$. In most practical scenarios, the latter is preferable, as it naturally allows *out-of-sample* estimation, i.e., provides GWOT mapping for new (unseen) samples $\mathbf{x} \sim \mathbb{P}$. In our paper, we deal with continuous setup.

## 3 Continuous Gromov-Wasserstein solvers

Here we outline the current progress in solving the GWOT problem specifically focusing on the continuous formulation. Most of the GWOT solvers are only discrete or adapted to *emulate a continuous behavior* by implementing some specific out-of-sample estimation method on top of the results of some discrete solver. The initial approach to solve the GWOT problem in discrete case (§2.2) was introduced in (Mémoli, 2011; Peyré et al., 2016). Below we only detail the methods which specifically aim to solve the continuous formulation and provide the out-of-sample estimation. The sanity-check of the solvers' implementations on a toy $\mathbb{R}^3 \to \mathbb{R}^2$ problem is available in Appendix B.

**StructuredGW (Sebbouh et al., 2024)**. It introduces an iterative algorithm for solving the discrete entropy-regularized inner product case of the GW problem with $p = 2$, using the reformulation from (Vayer, 2020, maxOT). At each step, the coupling matrix $\mathbf{T}$ is updated via Sinkhorn iterations, alongside an auxiliary rotation matrix is updated by various methods depending on the regularization. For out-of-sample estimation, the method uses entropic maps (Pooladian & Niles-Weed, 2024; Dumont et al., 2024), using a learned dual potential to perform the mapping.

**FlowGW (Klein et al., 2023).** The proposed framework consists in fitting a discrete GW solver inspired by (Peyré et al., 2016) to obtain a coupling matrix $\mathbf{T}$. This coupling matrix helps to figure out the best way to match available samples from source to target domains. Weighted pairs of source and target samples are constructed using the distribution described by the coupling matrix. Then these samples are used to train a Conditional Flow Matching (CFM) model (Lipman et al., 2023) with the noise outsourcing technique from (Kallenberg, 1997). The inference process is done by solving the ODE given by the CFM model.

**AlignGW (Alvarez-Melis & Jaakkola, 2018)**. This work proposes a discrete solver for the alignment of word embeddings. The authors use Sinkhorn iterations to compute the updates on the coupling

matrix instead of a linear search implemented in the Python Optimal Transport library (Flamary et al., 2021) that is inspired by (Titouan et al., 2019a; Peyré et al., 2016). This change significantly improves the stability of the solver. In spite of being a totally discrete method, we consider it due to its performance in challenging tasks. In order to allow out-of-sample estimations, we train a Multi-Layer Perceptron (MLP) model on the argmax based matrix from the learned coupling matrix $\mathbf{T}$.

**RegGW (Uscidda et al., 2024)**. This work extends the Monge Gap Regularizer (Uscidda & Cuturi, 2023) to the Gromov-Wasserstein setting by parameterizing the transport push-forward function $T$ with a neural network. The model is trained using a regularized loss that combines a *fitting* term—chosen as the Sinkhorn divergence (Uscidda et al., 2024)—to align the mapped points with the target distribution, and the Gromov-Monge gap regularizer, defined as the difference between the distortion of $T$ (i.e., its GW cost) and the true discrete GW solution. Unlike other methods such as (Sotiropoulou & Alvarez-Melis, 2024), this approach does not require access to an intermediate reference distribution, making it more practical for our experimental setting.

**CycleGW (Zhang et al., 2021)**. The authors propose minimizing the Unbalanced Bidirectional Gromov–Monge Divergence (UBGMD) to recover two push-forward maps: $f$ pushing $\mathbb{P}$ toward $\mathbb{Q}$, and $g$ doing the reverse. This is conceptually related to the Unbalanced Gromov-Wasserstein divergence Séjourné et al. (2020). Their approach involves minimizing a Generalized Maximum Mean Discrepancy (GMMD), computed using MMD with Gaussian kernels. In our setup, $f$ corresponds to our mapping function $T$. A related method from Hur et al. (2021) also incorporates an MMD term, but due to its similarity and lack of publicly available code, we focus on Zhang et al. (2021).

## 4    LIMITATIONS OF EXISTING METHODS

As mentioned before, the majority of existing Gromov-Wasserstein approaches rely on discrete GWOT formulation, see §2.2. Solving (2) becomes increasingly computationally expensive as the training sample sizes $N_x$ and $N_y$ grow. This renders some datasets to hardly be manageable by discrete solvers. However, in our paper, we specifically focus on the other sources of potential failures for GWOT, which stem from peculiarities of considered practical data setups. Below, we give a detailed description of this problem.

### 4.1    PITFALLS OF PRACTICAL DATA SETUP

To begin with, we introduce the notion of **statistical (un)relatedness** of data which undergoes Gromov-Wasserstein alignment. To fit a GWOT solver, a practitioner typically has two training datasets, $X = \{\mathbf{x}_i\}_{i=1}^{N_x}$ and $Y = \{\mathbf{y}_i\}_{i=1}^{N_y}$. They are sampled from the reference source ($\mathbb{P}$) and target ($\mathbb{Q}$) distributions, see our training setup, §2.3. In what follows, without loss of generality, we will assume $N_x = N_y \stackrel{\text{def}}{=} N$. The natural statistical assumption on samples $X$ and $Y$ is that they are mutually independent. We define this data setup as *statistically unrelated*. Simultaneously, we introduce an alternative setup under which the source and target datasets $X$ and $Y$ turn out to be connected by some specific statistical relationships. Let us assume that samples $X$ and $Y$ are generated according to a joint distribution $\pi \in \Pi(\mathbb{P}, \mathbb{Q})$, $\pi \neq \mathbb{P} \otimes \mathbb{Q}$. Practically, we expect that samples from this distribution are meaningfully dependent; i.e., $\pi$ is "significantly" different from the independent coupling $\mathbb{P} \otimes \mathbb{Q}$ due to the structural similarities present in the respective intra-domain geometries. In particular, coupling $\pi$ may even set a one-to-one correspondence between the domains. Then, we call the training samples $X$ and $Y$ to be *statistically related* if they are obtained with the following procedure:

1. First, we jointly sample $X$ and $\widetilde{Y} = \{\widetilde{\mathbf{y}}_i\}_{i=1}^N \subset \mathbb{R}^{d_y}$ from coupling $\pi$, i.e.: $X \times \widetilde{Y} \sim \pi$.
2. Secondly, we apply *an unknown permutation* $\sigma$ of indices to $\widetilde{Y}$ yielding $Y$, i.e.: $Y = \sigma \circ \widetilde{Y}$.

We found that the majority of the experimental test cases, on which the existing GWOT solvers are validated, frequently follow exactly the *statistically related* data setup. For instance, in the problem of learning cross-lingual word embedding correspondence (Alvarez-Melis & Jaakkola, 2018; Grave et al., 2019), the underlining coupling $\pi$ could be understood as the distribution of dictionary pairs. The other example is the bone marrow dataset (Luecken et al., 2021; Klein et al., 2023). In this case, the source and target samples are generated using two different methods to profile gene expressions on the same donors. Interestingly, the *statistically related* data setup suits well the discrete GWOT formulation, because the optimization problem in this case boils down to the search for the permutation $\sigma$ that spawned the target dataset $Y$. This leads to the natural **hypothesis** that for *statistically unrelated* training datasets $X$ and $Y$ the performance of existing GWOT solvers

may be poor. To test this hypothesis, we propose the experimental framework described in the next paragraphs.

**Modeling statistical (un)relatedness in practice.** Let $X = \{\mathbf{x}_1, \ldots, \mathbf{x}_N\} \sim \mathbb{P}$ and $Y = \{\mathbf{y}_1, \ldots, \mathbf{y}_N\} \sim \mathbb{Q}$ be the source and target datasets. We assume that $X$ and $Y$ are totally paired, i.e., every $i$-th vector in the source set $X$ is the pair of the $i$-th vector in the target set, $Y$. Also, we suppose that the pairing is reasonable, i.e., dictated by the nature of the data on hand. For example, if $X$ and $Y$ are word embeddings, then $\mathbf{x}_i$ and $\mathbf{y}_i$ correspond to the same word. We propose a way how to create training data with different levels of statistical relatedness. Initially, batches of $N$ paired (source and target) samples are randomly selected from the datasets, then split into train and test sets, $N = N_{train} + N_{test}$. The train samples are then divided into two halves and a value $\alpha$, $0 \leq \alpha \leq 1$ is set, this value represents the fraction of $N_{train}/2$ samples that will be paired.

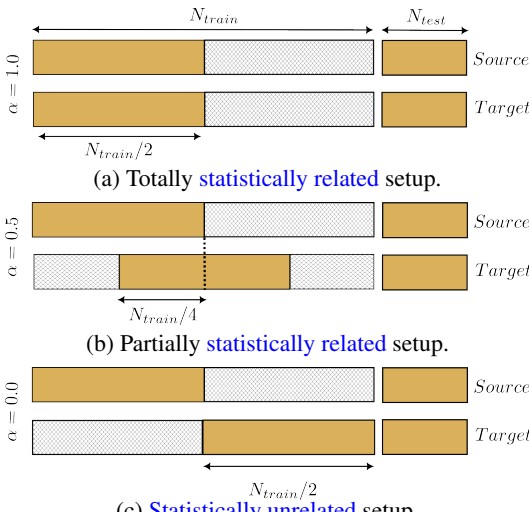

(a) Totally statistically related setup.

(b) Partially statistically related setup.

(c) Statistically unrelated setup.

Figure 2: Data splitting and statistical (un)relatedness.

The resulting training datasets (both source and target) will totally contain $N_{train}/2$ samples. They are formed by selecting specific indices from the original train sets. Indices from 0 to $N_{train}/2$ are taken from the source while indices from target are shifted, we take the $\lceil (1-\alpha)(N_{train}/2) \rceil$ to $\lceil (1-\alpha/2)N_{train} \rceil$ indices. Therefore, a value of $\alpha = 1$ will represent a *totally statistically related* setup (Figure 2a), $0 < \alpha < 1$ is *partially statistically related* (Figure 2b) and $\alpha = 0$ is *statistically unrelated* (Figure 2c).

It is important to emphasize that our benchmark construction requires datasets with (a) known ground-truth pairs, (b) alignable structures (ground-truth pairs are optimal pairs, i.e., they yield minimal distortion) and (c) large amount of samples to train continuous models.

To conclude the subsection, we want to emphasize that both *statistically related* and *statistically unrelated* setups are practically important. Some real-world use cases of the former include word embedding assignment and gene expression profiles matching problems, see the details in the text above. In turn, the *statistically related* setup naturally appears when aligning single-cell multi-omics data (Demetci et al., 2022). Here one aims at matching different single-cell assays, which are statistically unrelated, because applying multiple assays on the same single-cell is typically impossible.

### 4.2    BENCHMARKING GWOT SOLVERS ON STATISTICALLY (UN)RELATED DATA: GLOVE AND BPEMB EXPERIMENTS

In order to check how continuous GW solvers perform under statistically unrelated, partially and totally statistically related setups, we utilize two different text corpora: Twitter[1] and MUSE[2] (Multilingual Unsupervised and Supervised Embeddings) (Conneau et al., 2017) bilingual vocabularies. We then embed them using either the GloVe (Global Vectors for Word Representation) algorithm (Pennington et al., 2014) or BPEmb (Byte-Pair) (Heinzerling & Strube, 2018) embeddings. For now onward we refer as Twitter-GloVe and MUSE-BPEmb to the datasets of embeddings. Further details regarding the motivation to use these embeddings can be found in Appendix C.

For the Twitter-GloVe dataset, we define the *whole* data space as the first 400K word embeddings from a total of approximately 1.2 million. In the case of MUSE-BPEmb, the *whole* data space consists of 90K English-language samples. Results for the $100 \rightarrow 50$ dimensional setup appear in the main text, with additional configurations presented in Appendix D.

We test **three** baseline solvers introduced in § 3: StructuredGW, AlignGW, and FlowGW. We fit each solver for different values of $\alpha$ (levels of statistical relatedness), ranging from 0.0 to 1.0 in increments of 0.1. For each $\alpha$, we perform **ten** repetitions with different random seeds following the process

---

[1]https://radimrehurek.com/gensim/downloader.html
[2]https://github.com/facebookresearch/MUSE

described in § 4.1. We use $N_{\text{train}} = 6K$, i.e., each run uses source and target datasets containing $N_{\text{train}}/2 = 3K$ samples; $N_{\text{test}} = 2048$. The only reason for this relatively small training size is the computational complexity of the three solvers. The discrete optimization procedures underlying them cannot scale to significantly larger $N_{\text{train}}$. The training of RegGW and CycleGW baselines from § 3 fails with small datasets. These methods are therefore deferred to § 5.2, where larger datasets are considered. The results are shown in Figure 3. We refer to Appendix D for additional experimental setups.

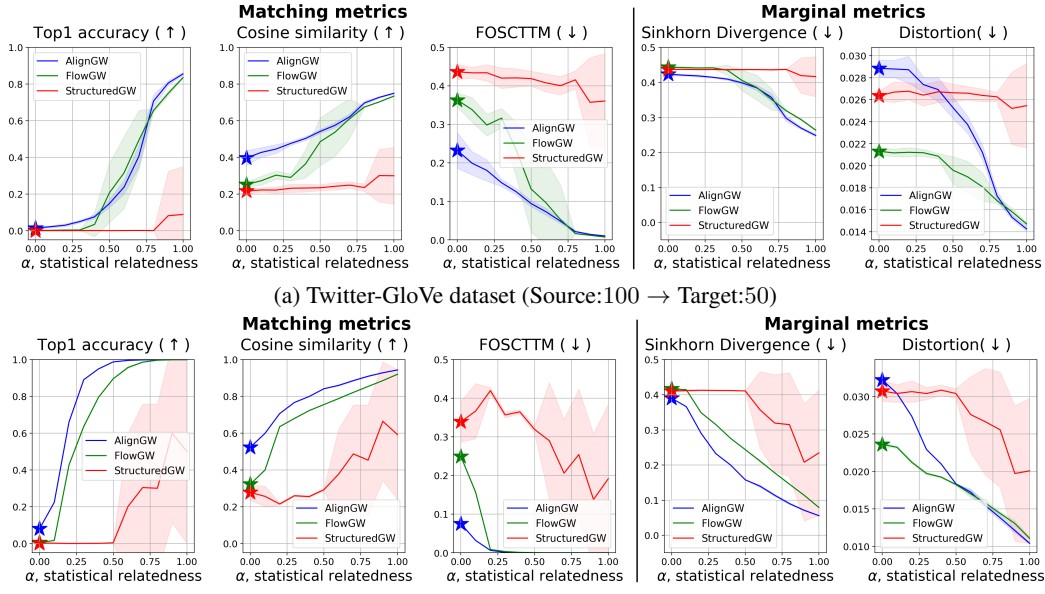

(a) Twitter-GloVe dataset (Source:100 → Target:50)

(b) MUSE-BPEmb (English) dataset (Source:100 → Target:50)

Figure 3: Performance of the baseline GWOT solvers for **Twitter-GloVe** (3a) and **MUSE-BPEmb (English)** (3b) at different levels of statistical relatedness $\alpha$ for the $100 \rightarrow 50$ setup. Solvers were trained with $N_{\text{train}}/2 = 3000$ samples from spaces of 400K and 90K, respectively; the plot shows results on a 2048-sample test subset. Accuracy metrics were computed over the combined *reference* space of $N_{\text{train}} + N_{\text{test}} = 8048$ samples.

For evaluation, we measured matching and marginal metrics; the main text reports matching metrics along with some marginal ones. As part of the matching metrics, we report Top-1 accuracy ↑, cosine similarity ↑, and FOSCTTM ↓. As marginal metrics we report Distortion and Sinkhorn divergence in the main text. Other additional marginal metrics can be found in Appendix D.1., a justification for the choice of matching and marginal metrics is provided in Appendix C.1. All metrics are computed on unseen test data, with *reference* points defined as the union of train and test sets, $N = N_{\text{train}} + N_{\text{test}} = 8048$.

**Conclusions.** The results indicate that almost all the baseline solvers perform well in totally statistically related scenarios, even when evaluated on unseen data. StructuredGW presents a less robust behavior. This demonstrates their ability to learn and capture the inner structures when the data is highly statistically related. However, their performance drops significantly as the value of $\alpha$ decreases. This finding clearly highlights a key limitation: current GWOT methods are restricted to work on very limited setups where source and target datasets are highly statistically related what is not practically common. Additionally, if high statistical relation exists, the GW alignment problem may be treated as a matching problem

**Towards solving pitfalls of existing GWOT solvers.** We conjecture that the observed behavior is mainly due to the small sizes of training sets dictated by the discrete nature of the solvers. Indeed, relatively small samples hardly could fully express the intrinsic geometry of the data, which complicates the faithful GW alignment of the domains, see Appendix F.1 for the broader discussion on the issue. Therefore, one possible way to increase the performance is to consider GW solvers adapted to large amount of data. In the following section (§5), we check different possibilities. In particular, we propose a new continuous solver (§5.1) which does not rely on discrete techniques. Therefore, it can better capture the inner geometry and structure of the data without the strict need of training on statistically related data as well as allowing training on large datasets.

**DISCLAIMER.** With the provided benchmark pipeline we do not aim to demonstrate that some GW solvers do not work. Instead, we assume that all considered GW solvers work properly if applied correctly. In particular, even under an statistically unrelated setup, see Figure 2, solvers may indeed accurately recover a GW coupling.

## 5 GWOT SOLVERS AT LARGE SCALE

In this section, we start by introducing NeuralGW, a novel scalable method to solve the continuous GWOT problem (§5.1). Then we proceed to the practical performance of NeuralGW and the baselines in large-scale GloVe benchmark (§5.2).

### 5.1 NEURAL GROMOV-WASSERSTEIN SOLVER

In this subsection, we conduct the theoretical and algorithmic derivation of our proposed approach. In what follows, we restrict to the case $d_x \geq d_y$; source ($\mathbb{P}$) and target ($\mathbb{Q}$) distributions are absolutely continuous and supported on some compact subsets $\mathcal{X} \subset \mathbb{R}^{d_x}$, $\mathcal{Y} \subset \mathbb{R}^{d_y}$ respectively. Our method is developed for innerGW, i.e., problem (3) with $c_{\mathcal{X}} = \langle \cdot, \cdot \rangle_{d_x}$, $c_{\mathcal{Y}} = \langle \cdot, \cdot \rangle_{d_y}$, $p = 2$:

$$\text{innerGW}_2^2(\mathbb{P}, \mathbb{Q}) \stackrel{\text{def}}{=} \min_{T_\sharp \mathbb{P} = \mathbb{Q}} \int_{\mathbb{R}^{d_x} \times \mathbb{R}^{d_x}} \left| \langle \mathbf{x}, \mathbf{x}' \rangle_{d_x} - \langle T(\mathbf{x}), T(\mathbf{x}') \rangle_{d_y} \right|^2 d\mathbb{P}(\mathbf{x}) d\mathbb{P}(\mathbf{x}'). \quad (4)$$

Note that the existence of minimizer for (4) is due to (Dumont et al., 2024, Theorem 3.2). We base our method on the theoretical insights about GW from (Vayer, 2020). According to (Vayer, 2020, Theorem 4.2.1), when $\int \|\mathbf{x}\|_2^4 d\mathbb{P}(\mathbf{x}) < +\infty$, $\int \|\mathbf{y}\|_2^4 d\mathbb{Q}(\mathbf{y}) < +\infty$, problem (4) is equivalent to

$$\text{innerGW}_2^2(\mathbb{P}, \mathbb{Q}) = \text{Const}(\mathbb{P}, \mathbb{Q}) - \max_{\pi \in \Pi(\mathbb{P}, \mathbb{Q})} \max_{P \in F_{d_x, d_y}} \int \langle \mathbf{P}\mathbf{x}, \mathbf{y} \rangle_{d_y} d\pi(\mathbf{x}, \mathbf{y}), \quad (5)$$

where $F_{d_x, d_y} \stackrel{\text{def}}{=} \{\mathbf{P} \in \mathbb{R}^{d_x \times d_y} \mid \|\mathbf{P}\|_{\mathcal{F}} = \min(\sqrt{d_x}, \sqrt{d_y})\}$ are the matrices of fixed Frobenius norm. Note that (5) admits a solution $\pi^* \in \Pi(\mathbb{P}, \mathbb{Q})$, $P^* \in F_{d_x, d_y}$ (Vayer, 2020, Lemmas 6.2.7; 4.2.2).

Our following lemma reformulates the innerGW problem as a minimax optimization problem. This reformulation is inspired by well-celebrated dual OT solvers such as (Korotin et al., 2021b; Fan et al., 2023; Korotin et al., 2023b).

**Lemma 5.1 (InnerGW as a minimax optimization)** *It holds that* (5) *is equivalent to*

$$\text{innerGW}_2^2(\mathbb{P}, \mathbb{Q}) = \text{Const}(\mathbb{P}, \mathbb{Q}) + \min_{P \in F_{d_x, d_y}} \max_{f: \mathbb{R}^{d_y} \to \mathbb{R}} \min_{T: \mathbb{R}^{d_x} \to \mathbb{R}^{d_y}} \mathcal{L}(\mathbf{P}, f, T); \quad (6)$$

$$\mathcal{L}(\mathbf{P}, f, T) \stackrel{\text{def}}{=} \int_{\mathbb{R}^{d_y}} f(\mathbf{y}) d\mathbb{Q}(\mathbf{y}) - \int_{\mathbb{R}^{d_x}} [\langle \mathbf{P}\mathbf{x}, T(\mathbf{x}) \rangle_{d_y} + f(T(\mathbf{x}))] d\mathbb{P}(\mathbf{x}).$$

The following theorem theoretically validates the minimax optimization framework for solving the GW problem. It shows that under certain conditions, the solution $T^*$ of the minimax problem (6) brings an optimal GW mapping.

**Theorem 5.2 (Optimal maps solve the minimax problem)** *Assume that there exists at least one GW map $T^*$. For any matrix $\mathbf{P}^*$ and any potential $f^*$ that solve* (6)*, i.e.,*

$$\mathbf{P}^* \in \underset{P \in F_{d_x, d_y}}{\text{argmin}} \max_{f} \min_{T: \mathbb{R}^{d_x} \to \mathbb{R}^{d_y}} \mathcal{L}(P, f, T),$$
$$f^* \in \underset{f}{\text{argmax}} \min_{T: \mathbb{R}^{d_x} \to \mathbb{R}^{d_y}} \mathcal{L}(\mathbf{P}^*, f, T),$$

*and for any GW map $T^*$, we have:*

$$T^* \in \underset{T: \mathbb{R}^{d_x} \to \mathbb{R}^{d_y}}{\text{argmin}} \mathcal{L}(\mathbf{P}^*, f^*, T). \quad (7)$$

To optimize 6 we follow the best practices from the field of continuous OT (Korotin et al., 2021c; Fan et al., 2023; Korotin et al., 2023a;b; Asadulaev et al., 2024; Choi et al., 2023; Gushchin et al., 2024) and simply parameterize $f$ and $T$ with neural networks. In turn, $\mathbf{P}$ is a learnable matrix of fixed Frobenius norm. We use the alternating stochastic gradient ascent/descent/ascent method to train their parameters. The learning algorithm is detailed in the Appendix E. We call the method NeuralGW. As the sanity check, we run our proposed approach on toy 3D→2D setup, see Figure 5b in Appendix B. Note that in comparison to other continuous GWOT approaches, our method *does not rely on discrete OT* in any form. In particular, the training process assumes access to just random samples from $\mathbb{P}, \mathbb{Q}$; it does not use/need any pairing between them.

## 5.2 PRACTICAL PERFORMANCE OF NEURALGW AND BASELINES AT LARGE SCALE

We start by introducing our **large-scale** Twitter-GloVe and MUSE-BPEmb setups. We consider the same data preparation process as in §4.2, but we respectively take $N_{train} = 380$K and $N_{train} = 90$K samples instead of the 6K used in §4.2 for the baseline solvers; $N_{test} = 2048$ is left the same. Therefore, each repetition consists in training the models with the same data (190K or 45K samples) but using different initialization parameters for, e.g., the neural networks. As the *reference* dataset for metrics computation, we used the *whole* datasets of Twitter-GloVe and MUSE-BPEmb embeddings (400K and 90K respectively).

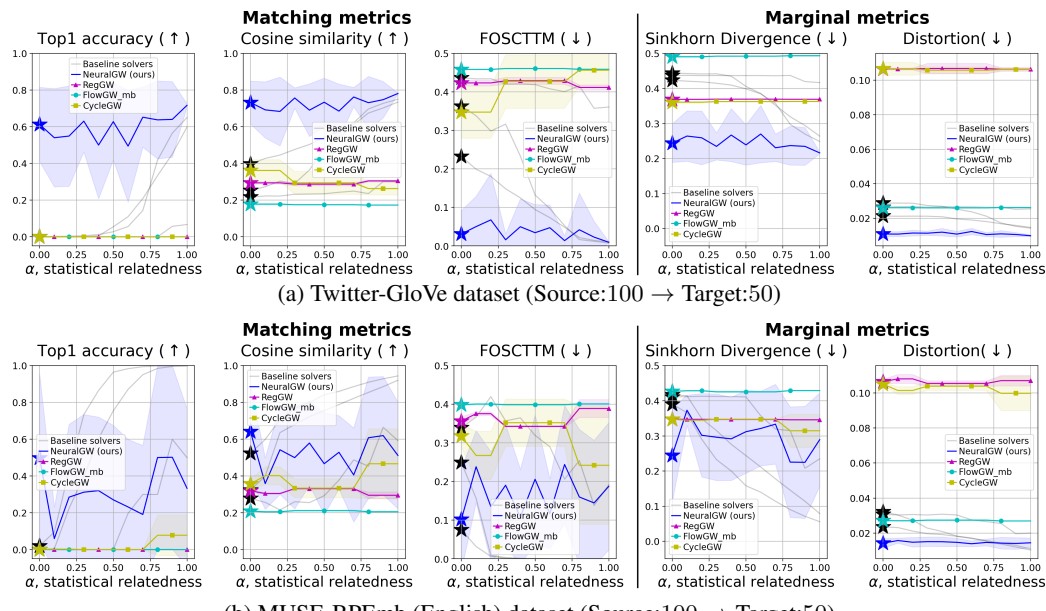

(a) Twitter-GloVe dataset (Source:100 → Target:50)

(b) MUSE-BPEmb (English) dataset (Source:100 → Target:50)

Figure 4: Performance of the baseline GWOT solvers for the **Twitter-GloVe** (4a) and **MUSE-BPEmb (English)** embeddings (4b) at different levels of statistical relatedness $\alpha$ for the 100 → 50 dimensionality setup. The solvers were trained with $N_{train}/2 = 380$K and $N_{train}/2 = 88$K samples, respectively. This plot shows results for a testing subset of 2048 samples, the accuracy metrics were computed considering *reference spaces* of 400K and 90K samples, respectively.

The competitive methods for the comparison under **large-scale** setups are: NeuralGW (§5.1); RegGW (§3); CycleGW (§3) and FlowGW (§3). The latter is trained in a minibatch manner, i.e., it fits flow matching on top of discrete GW solutions for minibatches. For completeness, we additionally report the performance of the baselines from §4.2 (in gray, labelled as "Baseline solvers", Figure 4). Note that, in this plot, they are trained on a small subset ($N_{train} = 6$K), but the metrics are computed with respect to the *whole reference* spaces, similar to NeuralGW, RegGW, CycleGW and FlowGW.

The colored charts for baselines' metrics computed in the whole Twitter-GloVe and MUSE-BPEmb *reference*) are available in Figure 8, Appendix D.4. While the comparison of the methods trained on 3K samples (baselines in gray) and 190K or 90K samples (our approach) might seem unfair, we stress that the sizes of datasets are selected based on the computational capabilities of the solvers. Based on the obtained results in Figure 4, we may notice the following:

(a) NeuralGW is the only method which may deal with large datasets for all levels of statistical relatedness. However, its performance is unstable across iterations.

(b) The advanced baselines (RegGW, CycleGW and FlowGW) failed to achieve reasonable performance for any value of $\alpha \leq 1$. This happens because, although the source and target datasets are globally statistically related, the random permutation applied to the target data (see §4.1) leads to batches with little or no statistical relation. Consequently, these models are trained on batches with $\alpha_{\text{batch}} \approx 0$, and due to their inherent reliance on discrete GW techniques, they are unable to learn a meaningful transport function.

(c) The choice of embedding used to generate the dataset directly impacts the performance of GW solvers and can lead to improved transport even at lower levels of statistical relatedness.

For the sake of completeness, we provide two additional experiments trained on Twitter-GloVe and MUSE-BPEmb (English): 1) NeuralGW trained on $3K$ samples (see Appendix D.4). The results are bad, which is expected because NeuralGW is based on complex adversarial procedure while the dataset is small. 2) RegGW, CycleGW and FlowGW (minibatch) trained on statistically related batches (see Appendix D.3) where each batch from the source is explicitly aligned with a statistically related batch from the target. This produces an improvement in performance; however, this setup is unfair due to this artificial construction, and the corresponding results should be interpreted with caution. We also provide marginal metrics for the setups in Figures 3 and 4, see Appendix D.1.

Additional Twitter-BPEmb results for the $100 \rightarrow 50$ setup are provided in Appendix D.5. As shown there, Twitter-BPEmb exhibits an interesting pattern: the performance of NeuralGW varies only slightly from Top@1 to Top@10. We further analyze and discuss this behavior in Appendix F.3.

**Conclusions.** Our proposed method (NeuralGW) is one of the first solver for the GWOT problem that does not rely on discrete approximations and hence can handle more challenging and realistic setups with statistically unrelated data. Specifically, we attract the readers' attention to metrics' values at $\alpha = 0$ which are highlighted with the star $\star$ symbol. In all the cases (Figure 4), our method outscores competitors by a large barrier. Our NeuralGW supports gradient ascent-descent batch training on large datasets. This capability enables the solver to learn intricate substructures even when trained on statistically unrelated batches/data. The initial results for our method suggest the potential to develop a general GWOT solver that is independent of data statistical relations, a significant advantage given that real-world datasets often lack such statistical relation.

In spite of achieving the best performance on statistically unrelated data, the results are inconsistent with respect to the initialization parameters, see a high standard deviation among repetitions. This inconsistency may be due to the minimax nature of the optimization problem. Additionally, adversarial methods like our NeuralGW are known to require large amounts of data for training, which can lead to issues when working with small datasets. We explore more general problems for baseline and NeuralGW solvers in Appendix D.

## 6 DISCUSSION

The general scope of our paper is conducting in-depth analyses of machine learning challenges that yield important new insights. In particular, we analyze Gromov-Wasserstein Optimal Transport solvers, identify the problems and propose some solutions. Our work clearly shows that while existing Gromov-Wasserstein Optimal Transport methods exhibit considerable success when solving downstream tasks, their performance may severely depend on the intrinsic properties of the training data. We partially address the issue by introducing our novel NeuralGW method. However, it has its own disadvantages. In particular, it is based on adversarial training which may be unstable and is not guaranteed to converge to an optimal solution of the GW problem. Thereby, our work witnesses that GWOT challenge in ML still awaits its hero who will manage to propose a reliable general-purpose method for tackling the problem.

**Reproducibility.** We provide the experimental details in Appendix E and the code to reproduce the conducted experiments in the supplementary material (see `readme.md`).

**LLM Usage.** Large Language Models (LLMs) were used only to assist with rephrasing sentences and improving the clarity of the text. All scientific content, results, and interpretations in this paper were developed solely by the authors.

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

CONTENTS

## A  Proofs of Theorems and Lemmas

**Proof of Lemma 5.1.** First, we recall the dual formulation of (1), see, e.g., (Fan et al., 2023):

$$\text{OT}(\mathbb{P}, \mathbb{Q}) \stackrel{\text{def}}{=} \max_f \left[ \min_T \int \big( c(\mathbf{x}, T(\mathbf{x})) - f(T(\mathbf{x})) \big) d\mathbb{P}(\mathbf{x}) + \int f(\mathbf{y}) d\mathbb{Q}(\mathbf{y}) \right], \tag{8}$$

respectively. Note that the existence of a solution $(f^*, T^*)$ of 8 is due to (Fan et al., 2023, Theorem 2). Our proof starts with (5). For each $\mathbf{P}$, we rewrite the inner optimization over $\mathbf{P}$ using (8) for the cost $c(\mathbf{x}, \mathbf{y}) = -\langle \mathbf{Px}, \mathbf{y} \rangle$:

$$\text{Const} (\mathbb{P}, \mathbb{Q}) - \text{innerGW}_2^2(\mathbb{P}, \mathbb{Q}) = \max_{\mathbf{P} \in F_{d_x, d_y}} \max_{\pi \in \Pi(\mathbb{P}, \mathbb{Q})} \int_{\mathbb{R}^{d_x} \times \mathbb{R}^{d_y}} \langle \mathbf{Px}, \mathbf{y} \rangle_n d\pi(\mathbf{x}, \mathbf{y}) =$$

$$- \min_{\mathbf{P} \in F_{m,n}} \left[ \min_{\pi \in \Pi(\mathbb{P}, \mathbb{Q})} \int_{\mathbb{R}^{d_x} \times \mathbb{R}^{d_y}} -\langle \mathbf{Px}, \mathbf{y} \rangle_{d_y} d\pi(\mathbf{x}, \mathbf{y}) \right] =$$

$$- \min_{\mathbf{P} \in F_{d_x, d_y}} \left[ \max_f \int_{\mathbb{R}^{d_x}} f(\mathbf{y}) d\mathbb{Q}(\mathbf{y}) + \min_{T \colon \mathbb{R}^{d_x} \to \mathbb{R}^{d_y}} \int_{\mathbb{R}^{d_x}} -\langle \mathbf{Px}, T(\mathbf{x}) \rangle_{d_y} - f(T(\mathbf{x})) d\mathbb{P}(\mathbf{x}) \right] =$$

$$- \min_{\mathbf{P} \in F_{d_x, d_y}} \max_f \min_T \mathcal{L}(\mathbf{P}, f, T).$$

**Proof of Theorem 5.2**. We expand $\mathcal{L}(\mathbf{P}^*, f^*, T^*)$ and use the fact that $T^*$ is the OT map.

$$\mathcal{L}(\mathbf{P}^*, f^*, T^*) = \int_{\mathbb{R}^{d_y}} f^*(\mathbf{y}) d\mathbb{Q}(\mathbf{y}) - \int_{\mathbb{R}^{d_x}} \langle \mathbf{P}^* \mathbf{x}, T^*(\mathbf{x}) \rangle_{d_y} + f^*\big(T^*(\mathbf{x})\big) d\mathbb{P}(\mathbf{x}). \tag{9}$$

Since $T^*$ is an OT map, we have $T^*_\sharp \mathbb{P} = \mathbb{Q}$, and by the change of variables formula we get:

$$\int_{\mathbb{R}^{d_x}} f^*\big(T^*(\mathbf{x})\big) d\mathbb{P}(\mathbf{x}) = \int_{\mathbb{R}^{d_y}} f^*(\mathbf{y}) d\mathbb{Q}(\mathbf{y}).$$

Plugging this into (9), we get:

$$\mathcal{L}(\mathbf{P}^*, f^*, T^*) = - \int_{\mathbb{R}^{d_x}} \langle \mathbf{P}^* \mathbf{x}, T^*(\mathbf{x}) \rangle_{d_y} d\mathbb{P}(\mathbf{x}).$$

Here, we once again use the fact that $T^*$ is the optimal transport map. Now, since $P^*$ and $f^*$ solve (4), we get the following:

$$\text{innerGW}_2^2(\mathbb{P}, \mathbb{Q}) = \text{Const}(\mathbb{P}, \mathbb{Q}) + \min_{T_\sharp \mathbb{P} = \mathbb{Q}} \mathcal{L}(\mathbf{P}^*, f^*, T)$$

Finally, from the fact that $\pi^* = [\text{id}_{\mathbb{R}^{d_x}}, T^*]_\sharp \mathbb{P}$ is optimal and (9), we have:

$$-\mathcal{L}(\mathbf{P}^*, f^*, T^*) = \int_{\mathbb{R}^{d_x} \times \mathbb{R}^{d_y}} \langle \mathbf{P}^* \mathbf{x}, \mathbf{y} \rangle_{d_y} d\pi^*(\mathbf{x}, \mathbf{y}) = \max_{\pi \in \Pi(\mathbb{P}, \mathbb{Q})} \int_{\mathbb{R}^{d_x} \times \mathbb{R}^{d_y}} \langle \mathbf{P}^* \mathbf{x}, \mathbf{y} \rangle_{d_y} d\pi(\mathbf{x}, \mathbf{y}) =$$

$$- \min_{T_\sharp \mathbb{P} = \mathbb{Q}} \mathcal{L}(\mathbf{P}^*, f^*, T^*),$$

which completes the proof.

## B  TOY SANITY-CHECK EXPERIMENT

To illustrate the capabilities of the solvers and as a necessary sanity check for the implementations, we propose a toy experiment. In this setup, the source distribution is a mixture of Gaussians in $\mathbb{R}^3$ and the target is also a mixture of Gaussians but in $\mathbb{R}^2$, see Figure 5a. By choosing this experiment on incomparable spaces, we ensure the solvers are actually capable of dealing with a real Gromov-Wasserstein problem. The obtained results for the baselines can be found in Figure 5d and 5c, Figure 5b shows the result for our method, NeuralGW. As we can see, for all methods a component of the source distribution is mostly mapped to a component/neighbouring components of the target distribution, indicating the correct GW alignment. We also report the distortion $d$ obtained after the methods were trained, the ground truth value for Figure 5a was computed using the discrete GW solver from the POT library.

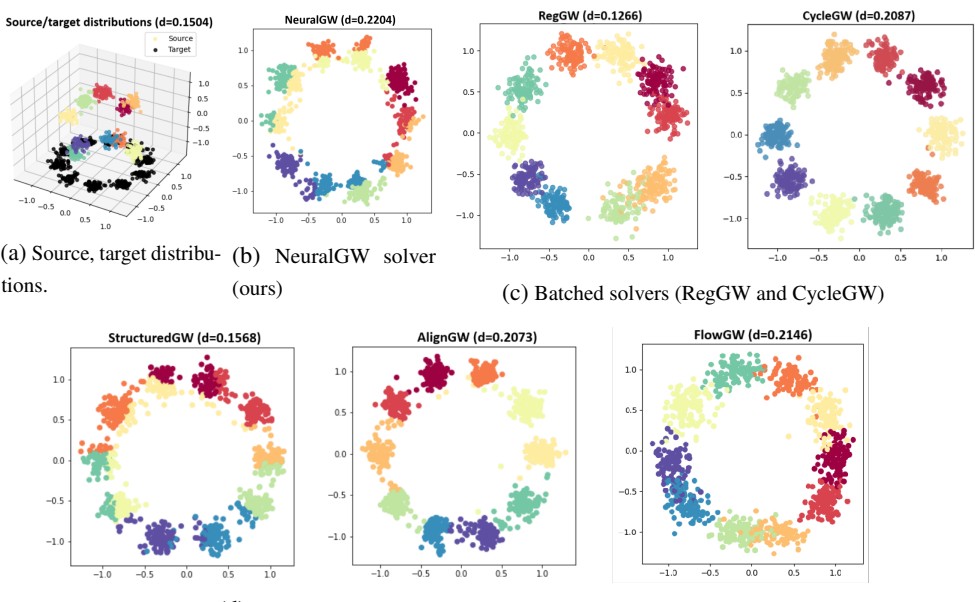

(a) Source, target distributions.

(b) NeuralGW solver (ours)

(c) Batched solvers (RegGW and CycleGW)

(d) Baseline solvers (StructuredGW, AlignGW, FlowGW).

Figure 5: Learned GWOT map $T$ by different solvers; Toy (3D→2D) experiment.

## C  METRICS AND EMBEDDINGS

### C.1  METRICS

This subsection explains how our evaluation metrics are computed. In the main text, we report what we refer to as *matching metrics*. These include Top-$k$ accuracy, cosine similarity, and the Fraction of Samples Closer Than the True Match (FOSCTTM). Additionally, in Appendix D, we report the *marginal metrics*, Maximum Mean Discrepancy (MMD) (Gretton et al., 2012), Sinkhorn divergence (Feydy et al., 2018), Bures-Wasserstein unexplained variance percentage ($\text{BW}_2^2$-UVP) (Korotin et al., 2021a) and distortion. All the metrics (with exception of FOSCTTM) are computed on normalized data, therefore, $\tilde{\mathbf{x}} = \mathbf{x}/\|\mathbf{x}\|_2$ where $\|\cdot\|_2$ denotes the Euclidean vector norm. Below we provide a concise explanation about how they are computed as well as the motivation to pick them.

**Matching metrics**

These metrics aim to provide an intuitive assessment of the quality of the predicted alignments by measuring how well the recovered correspondences match the ground truth. In order to compute them, it is necessary to know the actual ground-truth pairs.

- **Top $k$-accuracy** (↑). Let $\mathbf{x}_{\text{test}} \in \mathbb{R}^{m \times d_x}$ be the $m$ test samples from the source domain and $\mathbf{y}_{\text{pred}} \in \mathbb{R}^{m \times d_y}$ their predicted counterparts. For each $\hat{y}_j$ in $\mathbf{y}_{\text{pred}}$, we use `faiss` to retrieve the $k$ nearest neighbors from the reference space and denote their indices by $\{c_{j,1}, \ldots, c_{j,k}\}$. Given the

ground-truth labels $\{l_1, \ldots, l_m\}$ for the test samples, the Top-$k$ accuracy is defined as

$$\text{Top } k = \frac{1}{m} \sum_{j=1}^{m} \mathbb{1}\{ l_j \in \{c_{j,1}, \ldots, c_{j,k}\} \}.$$

The size of the *reference* space may vary depending on the specific experiment, as discussed in the main text. This metric is particularly useful due to its intuitive interpretability. Additionally, evaluating performance across different values of $k$ in Top-$k$ accuracy offers insight into how well the solver reconstructs the target distribution while preserving its internal structure.

- **Fraction of Samples Closed Than the True Match (FOSCTTM).** (↓) For each ground-truth pair $(x_i, y_i)$, let $\hat{y}_i = T(x_i)$ be the transported point and let $d(\cdot, \cdot)$ denote the Euclidean distance. We compute the proportion of target samples $y_j$ that satisfy $d(\hat{y}_i, y_j) < d(\hat{y}_i, y_i)$. Formally,

$$\text{FOSCTTM}(i) = \frac{1}{N} \left|\{j : d(\hat{y}_i, y_j) < d(\hat{y}_i, y_i)\}\right|.$$

The overall metric is the average $\text{FOSCTTM} = \frac{1}{N} \sum_{i=1}^{N} \text{FOSCTTM}(i)$. A perfect alignment yields $\text{FOSCTTM} = 0$. We note that this metric is insensitive to the choice of **reference** space, as it depends solely on distances to true paired samples.

- **Cosine similarity** (↑). It is computed between the predicted vector and the *reference* (optimal pair) vector in the target space.

**Marginal metrics**

We additionally report several metrics to ensure that the learned transport not only aligns individual samples but also accurately matches the overall distributions. The metrics considered are the Maximum Mean Discrepancy (MMD) (Gretton et al., 2012), the Sinkhorn divergence (Feydy et al., 2018), and the Bures-Wasserstein unexplained variance percentage (Korotin et al., 2021a). The metrics reported are computed for the testing data only. These metrics are computed between the empirical pushforward of the source distribution, $T_\sharp \hat{\mathbb{P}} \stackrel{\text{def}}{=} \frac{1}{N_x} \sum_{i=1}^{N_x} \delta(T(\mathbf{x}_i)) \approx \hat{\mathbb{Q}}$ (i.e., the output of the learned transport map or coupling) and the actual target empirical distribution, $\hat{\mathbb{Q}}$, to assess how well the learned mapping aligns the two distributions. However, these metrics do not directly reflect the quality of the solver itself, as they capture only distribution-level alignment and not, for example, matching or correspondence-based fidelity.

- **Maximum Mean Discrepancy (MMD)** (↓). It measures how different two distributions are by comparing their average representations in a high-dimensional feature space defined by a kernel. We denote $\mathbf{z} = T(\mathbf{x})$, therefore $\mathbf{z} \sim T_\sharp \mathbb{P}$. In our setup we used a Gaussian RBF kernel:

$$\text{MMD}^2(T_\sharp \mathbb{P}, \mathbb{Q}) = \mathbb{E}_{\mathbf{z}, \mathbf{z}' \sim T_\sharp \mathbb{P}}[k(\tilde{\mathbf{z}}, \tilde{\mathbf{z}}')] + \mathbb{E}_{\mathbf{y}, \mathbf{y}' \sim \mathbb{Q}}[k(\tilde{\mathbf{y}}, \tilde{\mathbf{y}}')] - 2\mathbb{E}_{\mathbf{z} \sim T_\sharp \mathbb{P}, \mathbf{y} \sim \mathbb{Q}}[k(\tilde{\mathbf{z}}, \tilde{\mathbf{y}})],$$

where $k(\mathbf{x}, \mathbf{y}) = \exp\left(-\frac{\|\mathbf{x} - \mathbf{y}\|^2}{2\sigma^2}\right)$, we set $\sigma = 10$.

- **Sinkhorn Divergence** (↓). It quantifies the discrepancy between two probability distributions, $T_\sharp \mathbb{P}$ and $\mathbb{Q}$ in our case.

$$S_\varepsilon(T_\sharp \mathbb{P}, \mathbb{Q}) = \text{OT}_\varepsilon(T_\sharp \mathbb{P}, \mathbb{Q}) - \frac{1}{2}\text{OT}_\varepsilon(T_\sharp \mathbb{P}, T_\sharp \mathbb{P}) - \frac{1}{2}\text{OT}_\varepsilon(\mathbb{Q}, \mathbb{Q}),$$

where $\text{OT}_\varepsilon$ is the entropy-regularized optimal transport cost with regularization strength $\varepsilon > 0$. In our computations we set $\varepsilon = 0.01$.

- **Bures-Wasserstein unexplained variance percentage ($\text{B}\mathbb{W}_2^2$-UVP)** (↓). The $\text{B}\mathbb{W}_2^2$-UVP metric measures how much of the target distribution's covariance is not captured by the pushforward of the source distribution.

$$\text{B}\mathbb{W}_2^2\text{-UVP}(T\sharp\mathbb{P}, \mathbb{Q}) = \frac{100\%}{\frac{1}{2}\text{Var}(\mathbb{Q})} \text{B}\mathbb{W}_2^2(T\sharp\mathbb{P}, \mathbb{Q})$$

where $\text{B}\mathbb{W}_2^2(T\sharp\mathbb{P}, \mathbb{Q})$ is the Bures–Wasserstein metric between the Gaussian approximations of the distributions $T\sharp\mathbb{P}$ and $\mathbb{Q}$.

- **Distortion** ($\downarrow$). This metric is related to the Gromov-Wasserstein cost introduced in 3.

$$\mathrm{D}_2^2(\mathbb{P}, T) \stackrel{\text{def}}{=} \int_{\mathbb{R}^{d_x} \times \mathbb{R}^{d_x}} (c_\mathcal{X}(\mathbf{x}, \mathbf{x}') - c_\mathcal{Y}(T(\mathbf{x}), T(\mathbf{x}')))^2 d\mathbb{P}(\mathbf{x}) d\mathbb{P}(\mathbf{x}').$$

In practice we compute it for the test samples according to the following expression:

$$\mathrm{D}(\tilde{\mathbf{x}}, T) \stackrel{\text{def}}{=} \frac{1}{n^2} \sum_{i,j=1}^n (c_\mathcal{X}(\tilde{\mathbf{x}}_i, \tilde{\mathbf{x}}_j) - c_\mathcal{Y}(T(\tilde{\mathbf{x}}_i), T(\tilde{\mathbf{x}}_j)))^2.$$

As this metric depends on the type of cost functions, we set $c_\mathcal{X} = c_\mathcal{Y} = c$ to be the cosine distance, $c(\tilde{\cdot}, \tilde{\cdot}') = 1 - \cos\text{-sim}(\tilde{\cdot}, \tilde{\cdot}')$. This choice of $c$, along with the normalization of input vectors to unit norm, ensures consistency across methods. In particular, approaches such as StructuredGW and NeuralGW employ the inner product as their intra-domain cost function, whereas others directly use the cosine distance. By normalizing and adopting a cosine-based cost, we unify the representation and make the comparisons between methods more meaningful.

### C.2 EMBEDDINGS

As mentioned in the main part of the paper, we use two different embeddings, GloVe and BPEmb, here we provide a more detailed explanation about how the datasets were constructed.

**GloVe embeddings and datasets**

The GloVe embeddings of words are generated using the GloVe algorithm (Pennington et al., 2014). Its main advantage is that the embedded vectors capture semantic relationships and exhibit linear substructures in the vector space. This allows meaningful computation of distances and alignments, which is central to the GWOT framework. The authors provide access to the GloVe embeddings of four datasets: Wikipedia, Gigaword, Common Crawl, and Twitter. We focus solely on the GloVe embeddings for the Twitter corpus. As we mentioned in the main text, we randomly take 400K samples from a total of 1.2 million, we use the pre-trained embeddings provided by the authors.

**Byte-Pair embeddings and datasets**

We explore the use of Byte-Pair embeddings (BPEmb), which is a subword tokenization method that breaks words into smaller units. It works by iteratively merging the most common pairs of adjacent symbols (like characters or character groups) in a corpus until a set vocabulary size is reached.

The MUSE embeddings were originally obtained using fastText (Joulin et al., 2016); however, most of the methods explored in this paper are unable to align these embeddings effectively. In the work we reference as the AlignGW solver (Alvarez-Melis & Jaakkola, 2018), the authors report positive alignment results on this dataset. However, it is important to note that these results may not fully reflect the true alignment capability of the method, as they incorporate cross-domain similarity local scaling (CSLS) (Conneau et al., 2017). While CSLS is commonly used in alignment tasks to enhance inference performance in multilingual translation, it introduces a corrective bias that may inflate the method's apparent success. This reliance on CSLS may thus lead to results that do not accurately represent the method's intrinsic alignment efficacy. In light of these considerations, we determined that a different approach was necessary for embedding the words from the MUSE vocabularies.

We generated the new dataset using the bpemb library[3], which provides pre-trained subword embeddings. For a thorough explanation of how these embeddings are derived, we recommend reviewing the original paper. To increase the chances that a word from the MUSE dictionaries appear in the BPEmbd vocabularies, we selected the largest available vocabulary size (200K) when loading the pre-trained embeddings. However, if a word still doesn't match, we chose to exclude them. Other reason to consider BPEmb is the possibility to compute the embeddings in different dimensions. For our experiments we only considered English and Spanish as bilingual dictionaries are provided for them. We excluded words with several translations and words with no translations. By doing this we ensure the obtained dataset of source and target embeddings fits our definition of statistical relatedness in Section §4.1.

---

[3]https://github.com/bheinzerling/bpemb/tree/master

With these considerations in mind, we constructed source and target datasets of BP embeddings for MUSE (English and Spanish) and Twitter corpora. Although the number of samples was reduced, the datasets remain viable for continuous methods.

# D  EXTENDED EXPERIMENTS

**Overview of the conducted experiments.** To help the reader navigating over all our considered experiments, We provide Table 1 summarizing the full list of experiments in our paper.

| Dataset name | Type of embedding | Size of the dataset | Train/test size | Source → Target | Evaluated on |
|---|---|---|---|---|---|
| Twitter | GloVe | 400K | Baseline solvers 6000/2048 | 100→50 | 8048 samples Figure 3 |
| | | | | | 400K samples Figure 8a |
| | | | | 50→25 | 400K samples Figure 10b |
| | | | | 50→100 | 400K samples Figure 10a |
| | | | | 25→50 | 400K samples Figure 10c |
| Twitter | GloVe | 400K | Continuous solvers 360K/2048 | 100→50 | 400K samples Figure 4a |
| | | | | 50→25 | 400K samples Figure 11b |
| | | | | 50→100 | 400K samples Figure 11a |
| Twitter | Byte-Pair | 90K | Baseline solvers 6000/2048 | 100→50 | 90K samples Figure 12 |
| | | | Continuous solvers 88K/2048 | 100→50 | |
| MUSE | Byte-Pair | 90K | Baseline solvers 6000/2048 | 100(English) → 50(English) | 90K samples Figure 8b |
| | | | Continuous solvers 88K/2048 | 100(English) → 50(English) | 90K samples Figure 4b |
| MUSE | Byte-Pair | 60K | Baseline solvers 6000/2048 | 100(English) → 100(Spanish) | 60K samples Table 3 |
| | | | Continuous solvers 58K/2048 | 100(English) → 100(Spanish) | |

Table 1: Summary of experiments

## D.1  MARGINAL METRICS FOR MAIN TEXT EXPERIMENTS

Here we report marginal metrics for our two main experiments: Twitter-GloVe and MUSE-BPEmb on baselines and continuous approaches. These new plots correspond to the matching metrics in Figures 3 and 4. The marginal metrics for the baselines can be found in Figure 6 and in Figure 7 for continuous solvers.

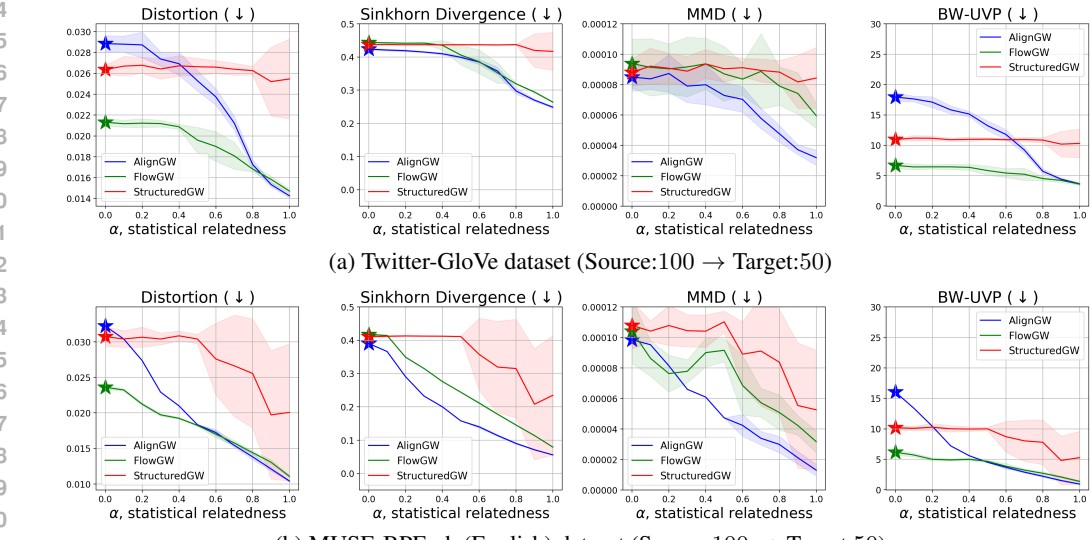

(a) Twitter-GloVe dataset (Source:100 → Target:50)

(b) MUSE-BPEmb (English) dataset (Source:100 → Target:50)

Figure 6: Marginal metrics for the baseline GWOT solvers for **Twitter-GloVe** (6a) and **MUSE-BPEmb (English)** (6b) at different levels of statistical relatedness $\alpha$ for the $100 \rightarrow 50$ setup. Solvers were trained with $N_{\text{train}}/2 = 3000$ samples from spaces of 400K and 90K, respectively; the plot shows results on a 2048-sample test subset. Accuracy metrics were computed over the combined *reference* space of $N_{\text{train}} + N_{\text{test}} = 8048$ samples.

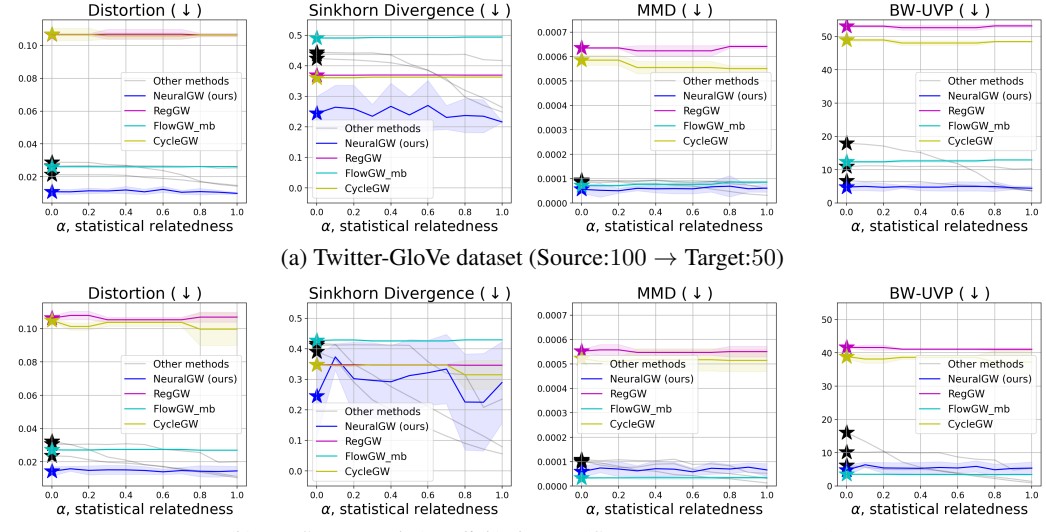

(a) Twitter-GloVe dataset (Source:100 → Target:50)

(b) MUSE-BPEmb (English) dataset (Source:100 → Target:50)

Figure 7: Marginal metrics for the continuous GWOT solvers for **Twitter-GloVe** (7a) and **MUSE-BPEmb (English)** (7b) at different levels of statistical relatedness $\alpha$ for the $100 \rightarrow 50$ setup. Solvers were trained with $N_{\text{train}}/2 = 3000$ samples from spaces of 400K and 90K, respectively; the plot shows results on a 2048-sample test subset. Accuracy metrics were computed over the combined *reference* space of $N_{\text{train}} + N_{\text{test}} = 8048$ samples.

**Conclusion.** We observe that marginal metrics correlate moderately with matching metrics, indicating that most methods align the marginal distributions to some extent. However, these metrics alone are insufficient to evaluate solver performance, as they ignore the structural alignment central to Gromov-Wasserstein objectives. While useful for assessing how target-like the transported samples are, marginal metrics should be considered complementary. Notably, our method, NeuralGW, consistently outperforms baselines in both structural and marginal alignment, and is the only approach capable of accurately matching marginals across all levels of statistical relation between source and target domains.

### D.2 TRAINED ON SMALL DATASET, EVALUATED W.R.T. LARGE *reference*

The accuracy evaluation involves identifying the $k$-nearest neighbours for a given vector within a target vector space. As discussed in Section §4.2, this space was limited to a small subset of $N_{train} + N_{test}$ samples for the experiments included in the main text, Figure 3. This is justified since the methods were trained on a similar number of samples. However, evaluating accuracy across the entire data space could offer deeper insights into the models' ability to capture the intrinsic structures of the probability distributions. Therefore, in Figure 8 we show the Top-$k$ accuracies computed with respect to the *whole reference space*. The other metrics are not affected.

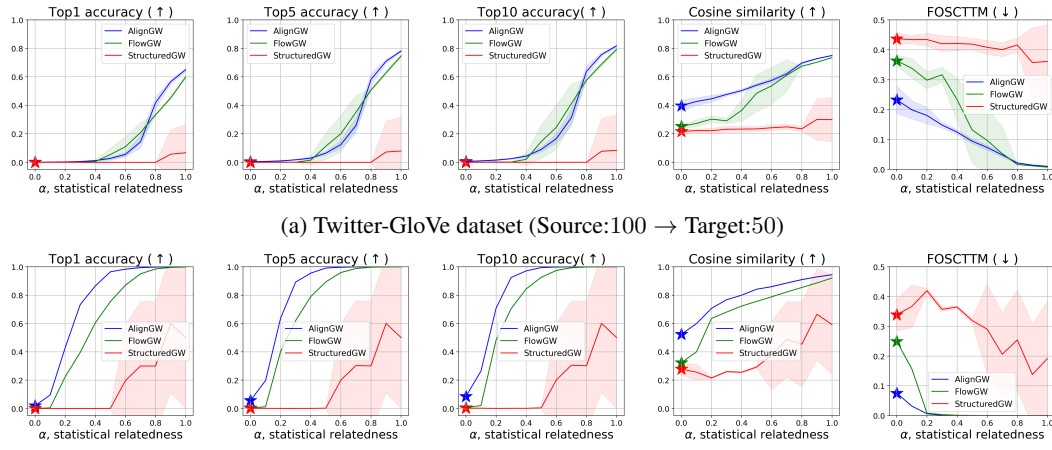

(a) Twitter-GloVe dataset (Source:100 → Target:50)

(b) MUSE-BPEmb (English) dataset (Source:100 → Target:50)

Figure 8: Performance of the baseline GWOT solvers for **Twitter-GloVe** (8a) and **MUSE-BPEmb (English)** (8b) at different levels of statistical relatedness $\alpha$ for the $100 \rightarrow 50$ setup. Solvers were trained with $N_{\text{train}}/2 = 3000$ samples from spaces of 400K and 90K, respectively; the plot shows results on a 2048-sample test subset. Accuracy metrics were computed over the *whole reference* space of 400K and 90K samples, respectively.

**Conclusion.** The performance of the baseline solvers drops significantly when computing the accuracies on the *whole reference* space. This decline is expected, as these methods are trained using a limited subset of the data, which restricts their ability to generalize effectively to unseen samples.

### D.3 EXPERIMENTS ON STATISTICALLY RELATED BATCHES FOR CONTINUOUS SOLVERS

As mentioned at the end of Subsection §5.2, we also report the performance of the continuous solvers when trained on statistically related batches under different levels of statistical relatedness $\alpha$. Figure 9 presents the results for one of our main experimental settings—Twitter-GloVe, from dimension 100 to 50—originally introduced in Figure 4a. The total number of training samples remains fixed at $N_{\text{train}} = 400K$, and we use batches of 500 samples across all methods. Further details on the hyperparameters are provided in Appendix E. We omit the MUSE-BPEmb case, as we expect qualitatively similar results.

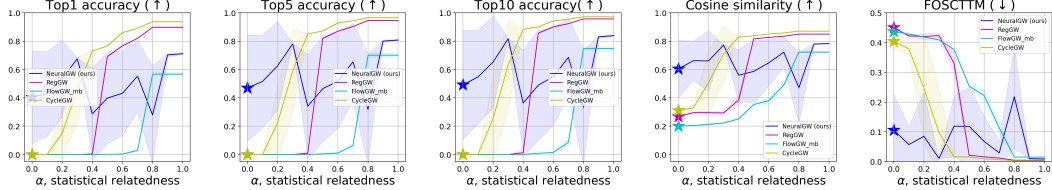

Figure 9: Matching metrics for continuous methods trained on **statistically related** **batches** on the Twitter-GloVe dataset (Source:100 → Target:50). $N_{train} = 400K$, $n_{batch} = 500$ and the Top-$k$ accuracy are computed on the *whole* vector space.

**Conclusion.** As mentioned in the main text, it is important to clarify that this setup is both unfair and unrealistic, as it requires providing statistically related batches to the solvers. We observe that this tweak significantly boosts performance compared to the fair setup, which confirms the correct implementation of the solvers and their proper functionality under specific, but artificial, conditions.

### D.4 GLOVE
**Experiments on additional dimensionalities for baseline and continuous solvers**

Figures 10 and 11 show the performance of the baseline and continuous methods, respectively, across additional dimensional configurations of the GloVe embedding. Specifically, we consider the transformations $50 \to 100$, $50 \to 25$, and $25 \to 50$. Accuracies are reported over the entire reference space, consisting of 400K samples. The parameters, training and test samples are the same as in the original experiments. For the continuous solvers we exclude the $25 \to 50$ setup as it produces a result similar to $50 \to 100$ due to the inability of the continuous solvers to transform low-to-high dimensional manifolds.

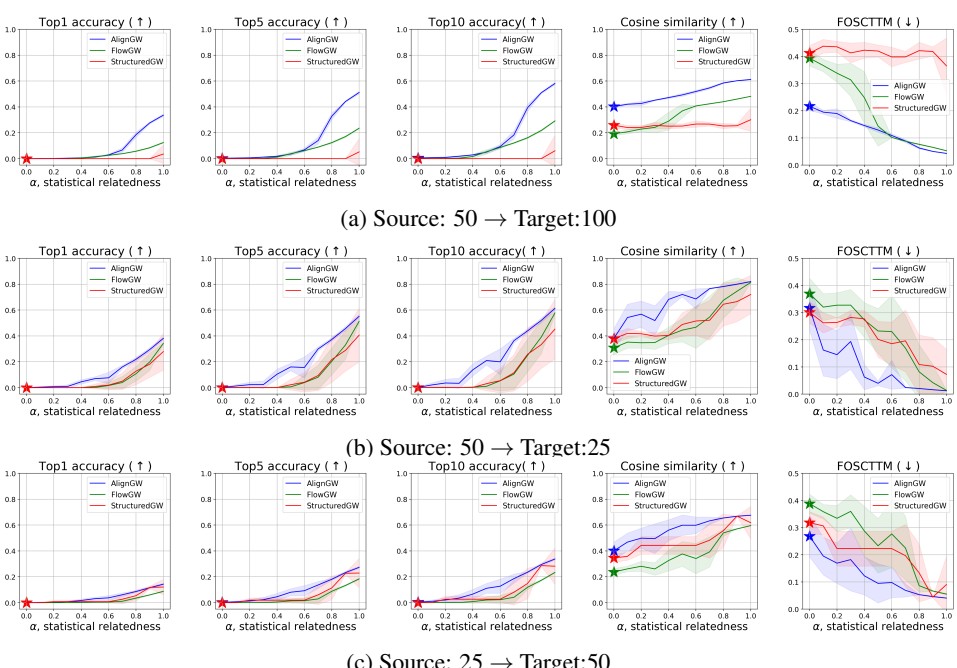

Figure 10: Performance of baseline GWOT solvers for the **Twitter-GloVe** embeddings at different statistical relatedness levels in additional dimensional setups. Accuracies computed on the *whole reference* space of 400K samples.

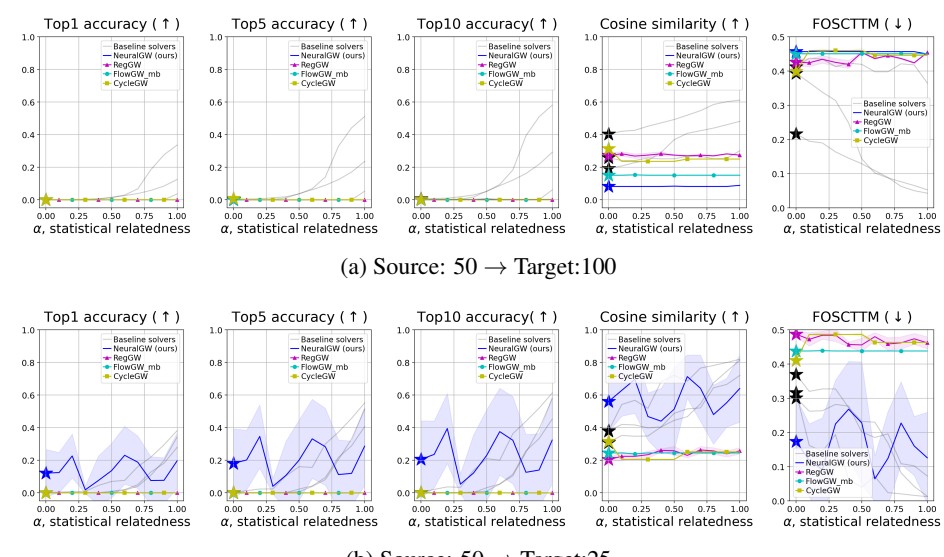

Figure 11: Performance of the continuous GWOT solvers for the **Twitter-GloVe** embeddings at different levels of statistical relatedness $\alpha$ in additional dimensional setups.

**Conclusion.** The performance of both baseline and continuous solvers drops significantly when the embedding dimensionality is reduced. This is expected, as lower-dimensional vectors may fail to adequately capture the structural features of the space. Conversely, none of the continuous solvers, including our NeuralGW, are able to effectively handle low-to-high dimensional setups.

**Additional experiments for NeuralGW for high-to-low dimension in small dataset ($N_{train} = 6K$).** Here we consider the same setup used for the baseline methods in §4.2 to show how a neural approach performs when it is trained using a limited amount of samples, $N_{train} = 6K$, see Table 2.

| Dimensions | Stat. Relatedness | Top 1 | Top 5 | Top 10 | Cosine similarity | FOSCTTM |
|---|---|---|---|---|---|---|
| | $\alpha = 0.2$ | 0.000 | 0.000 | 0.003 | 0.100 | 0.462 |
| $100 \rightarrow 50$ | $\alpha = 0.5$ | 0.000 | 0.001 | 0.004 | 0.107 | 0.470 |
| | $\alpha = 1.0$ | 0.000 | 0.003 | 0.005 | 0.117 | 0.447 |
| | $\alpha = 0.2$ | 0.000 | 0.001 | 0.003 | 0.165 | 0.441 |
| $50 \rightarrow 25$ | $\alpha = 0.5$ | 0.001 | 0.004 | 0.007 | 0.176 | 0.423 |
| | $\alpha = 1.0$ | 0.000 | 0.000 | 0.004 | 0.174 | 0.435 |
| | $\alpha = 0.2$ | 0.000 | 0.002 | 0.003 | 0.086 | 0.455 |
| $50 \rightarrow 100$ | $\alpha = 0.5$ | 0.000 | 0.003 | 0.005 | 0.084 | 0.459 |
| | $\alpha = 1.0$ | 0.000 | 0.002 | 0.003 | 0.089 | 0.450 |
| | $\alpha = 0.2$ | 0.000 | 0.002 | 0.005 | 0.113 | 0.440 |
| $25 \rightarrow 50$ | $\alpha = 0.5$ | 0.000 | 0.000 | 0.001 | 0.095 | 0.464 |
| | $\alpha = 1.0$ | 0.001 | 0.003 | 0.005 | 0.124 | 0.436 |

Table 2: Performance of the NeuralGW solver for the **Twitter-GloVe** embeddings at different levels of statistical relatedness $\alpha$ in **all** setups. The solver was trained with $N_{train}/2 = 3000$ samples from a whole space of 400K, this plot shows results for a testing subset of 2048 samples, the metrics were computed considering the whole 400K samples *reference* space.

**Conclusion.**

NeuralGW is unable to model the inner structures when the number of training samples is small (6K). In general, adversarial algorithms like those used to train NeuralGW require plenty of data to obtain meaningful results.

### D.5 BPEmb

**MUSE-BPemb, source: English (100) $\rightarrow$ target: Spanish (100)** We considered two different languages for source and target, the dimension of embedding was equal. The total number of samples was $N = 60K$, $N_{train} = 6K$ for baseline solvers and $N_{train} = 57K$ for NeuralGW, $N_{test} = 2048$ for both cases. The metrics were computed using the whole *reference* space. See Table 3 for the results.

| | **Baseline Solvers** | | | | | |
|---|---|---|---|---|---|---|
| | **FlowGW** | | **AlignGW** | | **StructuredGW** | |
| $\alpha$ | **Top 10** | **FOSCTTM** | **Top 10** | **FOSCTTM** | **Top 10** | **FOSCTTM** |
| **0.0** | 0.0013 | 0.4222 | 0.0012 | 0.4056 | 0.0000 | 0.5037 |
| **0.5** | 0.0029 | 0.3983 | 0.0022 | 0.3768 | 0.0006 | 0.4541 |
| **1.0** | 0.0267 | 0.3321 | 0.0146 | 0.3085 | 0.0009 | 0.4443 |
| | **Continuous solvers** | | | | | |
| | **RegGW** | | **FlowGW (mb)** | | **NeuralGW (ours)** | |
| $\alpha$ | **Top 10** | **FOSCTTM** | **Top 10** | **FOSCTTM** | **Top 10** mean (std) | **FOSCTTM** mean (std) |
| **0.0** | 0.0001 | 0.4521 | 0.0006 | 0.4767 | 0.0518 (0.0633) | 0.3811 (0.1286) |
| **0.5** | 0.0009 | 0.4411 | 0.0007 | 0.4770 | 0.0284 (0.0564) | 0.4379 (0.1067) |
| **1.0** | 0.0013 | 0.4384 | 0.0008 | 0.4772 | 0.0748 (0.0751) | 0.3514 (0.1321) |

Table 3: Results for MUSE dataset for English and Spanish as source and target languages, respectively, both are 100-dimensional BP embeddings.

**Conclusion.** We tested a selection of methods for this experiment, but none of them, whether baseline or continuous approaches, were able to produce successful results in this setup. It is worth noting

that the MUSE dataset was used in the original AlignGW paper; however, the type of embedding was different, resulting in a higher embedding dimensionality, which may have contributed to better performance in that context.

**Twitter dataset with different type of embedding:** For this case we considered the same dataset as for the GloVe experiments, but we changed the type of embedding to BPEmb, only the experiment for source: $100 \rightarrow$ target: 50 was performed. The total number of samples was $N = 90K$, $N_{train} = 6K$ for baseline solvers and $N_{train} = 87K$ for NeuralGW, $N_{test} = 2048$ for both cases. The metrics were computed as in the previous experiment. See Figure 12 for the results.

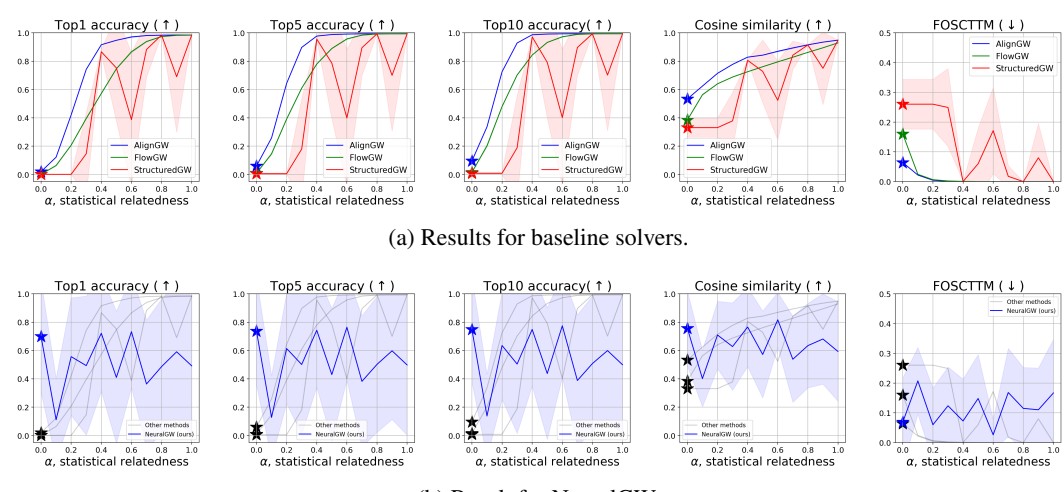

(a) Results for baseline solvers.

(b) Result for NeuralGW.

Figure 12: Performance of the baseline and batched GWOT solvers for the **Twitter-Byte-Pair** embeddings at different levels of statistical relatedness $\alpha$ in the $100 \rightarrow 50$ setup. **(a)** Baseline solvers trained with $N_{train}/2 = 3000$ samples from a whole space of 90K. **(b)** Batched solvers trained with $N_{train}/2 = 43K$ samples. In both cases the plot shows results for a testing subset of 2048 samples, the metrics were computed considering the whole 90K samples *reference* space.

For the sake of clarity, we only computed the metrics in the whole target space as in the second additional experiment in Appendix D.4. The number of training samples was kept $N_{train} = 6K$ for baseline solvers and $N_{train} = 87K$ for NeuralGW, $N_{test} = 2048$ for both cases.

**Conclusion.** The obtained results for the Twitter-BPEmb dataset clearly demonstrate the same trends as for GloVe embeddings (Sections 4.2 and 5.2 of the main text). For both MUSE and Twitter datasets, our NeuralGW turns out to be the only method capable to leverage statistically unrelated ($\alpha = 0$) setup when learning the map between embeddings of the same language but with different dimensions. At the same time, dealing with more complex *iter*-language setup turns out to be more difficult and none of the solvers have reached satisfactory quality on this setting. This underlines the inherent complexity of the GW problem.

### D.6 BIOLOGICAL DATASET

As seen in the previous experiments, the baselines and NeuralGW solvers have their own limitations and drawbacks heavily linked to their nature. However, in spite of their independent performance, they could partially recover the inner geometry of the domains. In this section, we propose a stress test scenario in which the solvers of the conventional GWOT problem yield poor results.

We explore the performance of the baselines solvers and NeuralGW in a biological dataset called bone marrow (Luecken et al., 2021) which is considered in the method that we named FlowGW (Klein et al., 2023). This dataset consists of 6224 samples of two different RNA profiling methods (ATAC+GEX and ADT+GEX), the samples in each domain belong to the same donors, therefore, the real pairs are known. The dimensionality for source and target domains are 38 and 50, respectively. Results can be found in Table 4. All the solvers are tested using 5000 samples for training and the rest for testing.

| | FlowGW | | AlignGW | | StructuredGW | | NeuralGW | |
|---|---|---|---|---|---|---|---|---|
| $\alpha$ | Top 10 | FOSCTTM | Top 10 | FOSCTTM | Top 10 | FOSCTTM | Top 10 | FOSCTTM |
| **0.2** | 0.004 | 0.486 | 0.004 | 0.488 | 0.001 | 0.489 | 0.003 | 0.479 |
| **0.5** | 0.004 | 0.489 | 0.004 | 0.483 | 0.004 | 0.487 | 0.003 | 0.514 |
| **1.0** | 0.004 | 0.49 | 0.004 | 0.484 | 0.004 | 0.49 | 0.003 | 0.459 |

Table 4: Results for bone marrow dataset.

**Conclusion.** In all the cases, the solvers could not properly replicate the inner geometry of the distributions even for totally statistically related setups, this leaded to get metrics corresponding to random guessing, i.e. accuracies close to 0 and FOSCTTM close to 0.5. There is one case of success for a solver trained on this dataset which corresponds to FlowGW (Klein et al., 2023), however, the reported results in their paper were obtained using a fused-GW solver instead of a conventional GW.

Finally, we can state that there is no general solver for the GWOT problem, all the currently available methods struggle when dealing with real world scenarios, i.e. statistically unrelated data, or with real world datasets, i.e. not consistent inner structures.

# E   SOLVERS' IMPLEMENTATION DETAILS

All the experiments were done without any normalization for the source and target vectors and for all the studied methods (baselines and NeuralGW). A total of ten repetitions were performed. It is important to clarify that the parameters listed below are the ones we used to align the embeddings, they may require some tweaks to make them work in the toy setup.

**StructuredGW.(Sebbouh et al., 2024)** We used the code from the official repository:

```
https://github.com/othmanesebbouh/prox_rot_aistats
```

As specified in Section §3, the algorithm uses an iterative solver that updates the cost matrix $\mathbf{T}$ by implementing several methods depending on the type of regularization, we only use the exact computation without any regularization. The plan $\pi$ is also updated every iteration by performing Sinkhorn iterations, we set this number of iterations to 1000. The entropy is set to $\varepsilon = 1e\text{-}4$. The total number of iterations is set to 200 or until convergence.

In this implementation, the authors use the Optimal Transport Tools (OTT) library (Cuturi et al., 2022). The computation time per repetition until convergence was 30 minutes in average on a CPU.

**AlignGW.(Alvarez-Melis & Jaakkola, 2018)** We use the official implementation of the method taken from the repository:

```
https://github.com/dmelis/otalign
```

We set the entropy term to $\varepsilon =$ 1e-3 and use the cosine distance to compute the source and target intra-cost matrices $\mathbf{C}^x$ and $\mathbf{C}^y$. We later normalize them by dividing them by their respective means as proposed in the original implementation. The model was trained on a CPU and the average training time was 10 minutes per repeat. The implementation uses the POT library. To get the predicted target from the obtained couplig $\mathbf{T}$, we first compute the indices `idx=np.argmax(T, axis=1)` and then the chosen target samples are `y_pred=y[idx]`. We chose this strategy as it gave better performance. Then, we train a `scikit-learn`'s MLPRegressor on top of this prediction as an inference method for test data, the parameters are: hidden_layer_sizes=256, random_state=1, max_iter=500.

**FlowGW. (Klein et al., 2023)** We used the implementation provided in the OTT library for the GENOT with slight modifications to adapt it to our pipeline. The hyperparameters for the vector field were as follows: Number of frequencies: 128, layers per block: 8, hidden dimension: 1024, activation function: SiLu, optimizer: AdamW (lr=1e-4). The Gromov-Wasserstein solver was set to work with entropy $\varepsilon =$ 1e-3 and using cosine distance to compute the intra-domain matrices.

**RegGW. (Uscidda et al., 2024)** For the sake of fairness, our implementation of this solver is based on the publicly available implementation for the Monge gap regularizer from the OTT library (Cuturi et al., 2022). To compute the Gromov-Wasserstein distance we used the GW solver from the library and took the entropy regularized cost. The following parameters were used for training: $\varepsilon_{fit} = 0.01$, $\varepsilon_{reg} = 0.001$, $\lambda = 1$. The transport model was parametrized as an MLP with $[512, 256, 256]$ dimensions for the hidden layers, the optimizer learning rate was 1e-4 and a batch size of 256.

**CycleGW (Zhang et al., 2021)**. We used the code provided by the authors in their repository:

https://github.com/ZhengxinZh/GMMD

As in the original implementation, we used fully connected neural networks (FCNN) to parametrize $f$ and $g$, in both cases the network consisted on a single layer with 512 neurons, and trained using the Adam optimizer with a learning rate of $1e$-3 (as suggested in the original paper). Both regularization parameters, $\lambda_x$ and $\lambda_y$ in the original paper, were set to 0.1. The multiplier of the distortion term was set to 5e-4. In spite of following the original implementation, it was not possible to make the solver work for our setups beyond the toy experiment.

### E.1 NEURALGW.

The innerGW problem in (4) can be optimized using our Algorithm 1.

---

**Algorithm 1:** Training algorithm for Neural Gromov-Wasserstein OT

---

1 **Input:** Distributions $\mathbb{P}$ and $\mathbb{Q}$ obtained from samples.
2 **Output:** Optimal rotation matrix $P_\omega$, critic $f_\theta$ and transport map $T_\gamma$.
3 **for** $i = 1, 2, 3, \ldots, n_{epochs}$ **do**
4  Sample batch from source and target distributions $X \sim \mathbb{P}, Y \sim \mathbb{Q}$.
5  **for** $i = 1, 2, 3, \ldots, k_P$ **do**
6   Compute $P$ loss $\mathcal{L}_P = -\frac{1}{N} \sum_{n=1}^{N} \langle \mathbf{P}_\omega \mathbf{x}, T_\gamma(\mathbf{x}) \rangle$
7   Gradient step over $\omega$ using $\frac{\partial \mathcal{L}_P}{\partial \omega}$
8   **for** $j = 1, 2, 3, \ldots, k_f$ **do**
9    **for** $k = 1, 2, 3, \ldots, k_T$ **do**
10     Compute mover loss $\mathcal{L}_T = -\frac{1}{N} \sum_{n=1}^{N} \langle \mathbf{P}_\omega \mathbf{x}, T_\gamma(\mathbf{x}) \rangle - \frac{1}{N} \sum_{n=1}^{N} f_\theta(T(\mathbf{x}))$
11     Gradient step over $\gamma$ using $\frac{\partial \mathcal{L}_T}{\partial \gamma}$
12    Compute critic loss $\mathcal{L}_f = -\frac{1}{N} \sum_{n=1}^{N} f_\theta(T_\gamma(\mathbf{x})) - \frac{1}{N} \sum_{n=1}^{N} f_\theta(\mathbf{y})$
13    Gradient step over $\theta$ using $\frac{\partial \mathcal{L}_c}{\partial \theta}$

---

Every experiment runs for 100 epochs. Each epoch iterates over the whole dataset (400K, 90K or 6K samples). $f$ and $T$ are parametrized using multi-layer perceptrons with $n_l$ with width $h$. Matrix $\mathbf{P}$ is parameterized by a single-layer linear neural network, we use GeoTorch Lezcano-Casado (2019) to constrain each row of its rows to lie on the unit sphere, ensuring row-wise normalization is maintained throughout training. These models are trained for $k_f, k_T$ and $k_p$ iterations, respectively. These parameters were chosen following the inner structure of the min-max-min problem, we optimized the inner transport model $T_\gamma$ more often than the others, from here we fine-tuned the values to get good metrics. The implementation details can be seen in Table 5. The batch size is 512 for the experiments with 400K and 90K samples and 64 for the experiments with 6K. Our code is written in `PyTorch`. Every epoch takes around to 3 minutes running on a GPU NVIDIA Tesla V100.

Table 5: Parameters for NeuralGW.

| | Model | $k$ | $n_l$ | $h$ | lr |
|---|---|---|---|---|---|
| 100→50 | $f$ | $k_f = 1$ | | | |
| 50→25 | $T$ | $k_T = 10$ | 4 | 512 | 1e-4 |
| 25→50 | $P$ | $k_P = 1$ | | | |
| | $f$ | $k_f = 1$ | | | |
| 50→25 | $T$ | $k_T = 10$ | 4 | 256 | 1e-4 |
| | $P$ | $k_P = 1$ | | | |

The loss for the critic $f$, i.e. $\mathcal{L}_f$ can use the well-known $R_1$ regularization Mescheder et al. (2018) adopted from GAN-like training methods. This regularized is used to stabilize our NeuralGW solver while training on word embeddings, the updated loss would be

$$\mathcal{L}_f = -\frac{1}{N} \sum_{n=1}^{N} f_\theta(T_\gamma(\mathbf{x})) - \frac{1}{N} \sum_{n=1}^{N} f_\theta(\mathbf{y}) - \lambda \frac{1}{N} \sum_{i=1}^{N} \left[ \left\| \nabla_{\mathbf{y}_i} f_\theta(\mathbf{y}_i) \right\|_2^2 \right],$$

with $\lambda = 0.1$. All the results for NeuralGW on word embeddings use this regularizer for training the critic.

## F BROADER DISCUSSIONS

### F.1 DISCRETE GW SOLVERS UNDER LOW STATISTICAL RELATEDNESS DATA SCENARIO

Our experimental results (Section 4.2 of the main text) testify that the discrete baseline solvers perform unsatisfactory when the level of statistical relatedness $\alpha$ in the data tends to zero. We hypothesis that the main reason behind this behavior is as follows. Small amount of data used for discrete solvers hardly could "catch" the intrinsic geometry of the underlining distribution. When we apply discrete GW solver, the Gromov-Wasserstein mapping is learned between the geometries induced by sample distributions, not original distributions, see Figure 13 for illustration. These

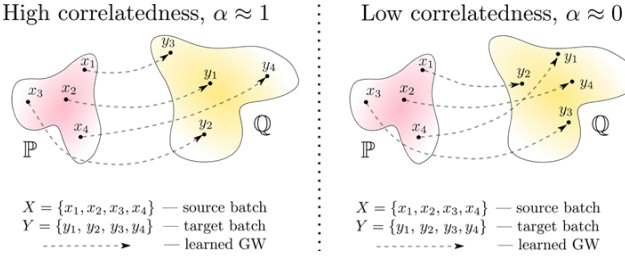

Figure 13: Discrete GW maps fitted under high (left) and low (right) levels of statistical relatedness.

"induced" geometries may be different from the original ones, they may have other symmetries and other properties. Matching them may result in Gromov-Wasserstein map which is quite different from the real GW map. Importantly, our hypothesis above is (indirectly) supported by recently derived sample complexity rates for Gromov-Wasserstein distance (Zhang et al., 2024). The authors consider GWOT problem (3) for quadratic costs $c_{\mathcal{X}}, c_{\mathcal{Y}}$. In this case, they observe that the sample convergence rate of the GW distance between empirical (sample-based) measures to the GW distance between reference measures $(\mathbb{P}, \mathbb{Q})$ is of the order $N^{\frac{-2}{\min d_{\mathcal{X}}, d_{\mathcal{Y}}}}$, where $N$ is the number of samples. Note that this discouraging exponential rate corresponds exactly to statistically unrelated ($\alpha \approx 0$) setup, where the discrete samples from the source and target reference distributions are independent. Besides, the sample complexity of *GW maps*, not distances, may be even worse.

On the other hand, when the level of statistical relatedness is high ($\alpha = 1$), the GW problem is reduced to finding the proper permutation of the data, see Figure 13, left part. The true solution of discrete GW in this case *coincides* with the true underlining GW map. If the learned map properly generalizes to new (unseen) samples, then the resulting performance is expected to be satisfactory. It is the behaviour we observe in our experiments, Section 4.2.

### F.2 ATTEMPTS TO STABILIZE NEURALGW

Here we provide a summary on our attempts to stabilize the training of NeuralGW.

**Input Convex Neural Networks (ICNN) as potential $f$.** Theoretically, the use of ICNN to parameterize the potential $f$ should improve the performance of the solver. However; the implementation of this parameterization as part of the NeuralGW solver provided not good results as well as an increased training time.

**Cayley transform to parameterize the matrix $P$.** We used a modified version of the Cayley transform to ensure the orthogonality of the matrix $P$, this allows us to have more control over the eigenvalues of the matrix, however, it also required more parameters to set. The results obtained were worse than the ones in the paper and the stability issue was not fixed.

**Data projection and normalization**. We also trained the solver on PCA and random Gaussian projections without any positive results. We obtained a similar outcome when training on normalized data.

**Larger batch sizes**. We trained using large batch sizes (i.e. 1024, 2048). This approach aims to better capture the cluster structure of the data by taking more samples at each training iteration. A slight improvement was provided, but it is not a robust choice across all setups.

### F.3 STUDY OF GLOVE VS. BPEMB

Here we provide a comparison between the Twitter-GloVE and BPEmb datasets for the $100 \rightarrow 50$ setup. We ran five runs for each $\alpha$. These experiments raised due to the interesting behavior observed in Figure 12. In these plots, Top-1 and Top-10 seem to have almost the same value. To understand the reason behind this behavior we propose to compute the accuracies for different values of $k$ for two different embeddings on the same dataset. As reference solvers we picked AlignGW and NeuralGW.

**AlignGW.**

As observed in Table 6, for BPEmb with higher $\alpha$, we observe near-perfect alignment already at very small $k$, so the Top@k saturates quickly to an accuracy of $1.0$. For GloVe with $\alpha = \{1.0, 0.5\}$, accuracy improves steadily with $k$, what is a regular behavior. The only configurations showing an unexpectedly slow improvement with respect to $k$ are GloVe and BPEmb for $\alpha = 0.0$. This corresponds to a **statistically unrelated** setting, where weak performance is naturally expected on Baseline solvers.

| Dataset | alpha | Top@1 | Top@10 | Top@20 | Top@50 | Top@500 | Top@2000 |
|---------|-------|-------|--------|--------|--------|---------|----------|
| Twitter GloVE | 1.0 | $0.65 \pm 0.01$ | $0.82 \pm 0.01$ | $0.85 \pm 0.01$ | $0.89 \pm 0.01$ | $0.95 \pm 0.00$ | $0.98 \pm 0.00$ |
| | 0.5 | $0.02 \pm 0.01$ | $0.08 \pm 0.02$ | $0.11 \pm 0.03$ | $0.16 \pm 0.03$ | $0.37 \pm 0.05$ | $0.55 \pm 0.04$ |
| | 0.0 | $0.00 \pm 0.00$ | $0.01 \pm 0.00$ | $0.01 \pm 0.00$ | $0.02 \pm 0.01$ | $0.08 \pm 0.02$ | $0.18 \pm 0.03$ |
| Twitter BPEmb | 1.0 | $0.98 \pm 0.40$ | $0.99 \pm 0.00$ | $1.00 \pm 0.00$ | $1.00 \pm 0.00$ | $1.00 \pm 0.00$ | $1.00 \pm 0.00$ |
| | 0.5 | $0.95 \pm 0.00$ | $0.99 \pm 0.00$ | $0.99 \pm 0.00$ | $1.00 \pm 0.00$ | $1.00 \pm 0.00$ | $1.00 \pm 0.00$ |
| | 0.0 | $0.03 \pm 0.00$ | $0.13 \pm 0.01$ | $0.17 \pm 0.01$ | $0.26 \pm 0.01$ | $0.55 \pm 0.03$ | $0.76 \pm 0.02$ |

Table 6: Top@k accuracies for different embeddings and alpha values for the AlignGW solver.

**NeuralGW.**

For GloVe embedding, accuracies increase clearly with $k$, showing that true correspondences are spread across many ranks rather than concentrated only at Top@1. The cases that show a small gap between Top@1 and Top@10 are BPEmb for $\alpha = [1.0, 0.0]$. In these settings, Top@k changes only slightly with respect to $k$. However, a closer look shows that this is caused by the **unstable performance** of NeuralGW in these configurations. As shown in Table 7, the standard deviation is very high in both cases. This is particularly important since for BPEmb some runs fitted the target space almost perfectly (Top@1 = 1) while others produced nearly no correct matches. This instability is what leads to the small differences observed between Top@1 and Top@10. Importantly, the case $\alpha = 0.5$, where the runs were stable, shows a much stronger sensitivity to $k$.

| Dataset | $\alpha$ | Top@1 | Top@10 | Top@20 | Top@50 | Top@500 | Top@2000 |
|---------|----------|-------|--------|--------|--------|---------|----------|
| Twitter GloVE | 1.0 | $0.57 \pm 0.28$ | $0.67 \pm 0.33$ | $0.69 \pm 0.34$ | $0.72 \pm 0.36$ | $0.76 \pm 0.37$ | $0.79 \pm 0.37$ |
| | 0.5 | $0.67 \pm 0.01$ | $0.81 \pm 0.01$ | $0.84 \pm 0.01$ | $0.88 \pm 0.01$ | $0.95 \pm 0.00$ | $0.97 \pm 0.00$ |
| | 0.0 | $0.64 \pm 0.00$ | $0.80 \pm 0.01$ | $0.83 \pm 0.01$ | $0.87 \pm 0.00$ | $0.94 \pm 0.00$ | $0.97 \pm 0.00$ |
| Twitter BPEmb | 1.0 | $0.39 \pm 0.48$ | $0.40 \pm 0.49$ | $0.40 \pm 0.48$ | $0.41 \pm 0.48$ | $0.43 \pm 0.47$ | $0.47 \pm 0.43$ |
| | 0.5 | $0.79 \pm 0.38$ | $0.81 \pm 0.36$ | $0.82 \pm 0.35$ | $0.83 \pm 0.33$ | $0.88 \pm 0.25$ | $0.91 \pm 0.18$ |
| | 0.0 | $0.20 \pm 0.39$ | $0.22 \pm 0.39$ | $0.23 \pm 0.38$ | $0.25 \pm 0.38$ | $0.34 \pm 0.35$ | $0.43 \pm 0.33$ |

Table 7: Top@k accuracies for different embeddings and alpha values for the NeuralGW solver.

