# OpenReview forum: "Uncovering Challenges of Solving the Continuous Gromov-Wasserstein Problem"
_ICLR.cc/2026/Conference — Submitted to ICLR 2026_

### Official Review · Reviewer_wskS · 2025-10-28

**Soundness:** 4
**Presentation:** 4
**Contribution:** 3
**Rating:** 8
**Confidence:** 4

**Summary:**

In this paper the authors investigate the challenges of solving continuous
Gromov-Wasserstein (GW) problems from discrete high dimensional data. The
authors first do a quick review of the existing literature on GW problems and
highlight the different strategies using discrete+regression or Neural networks
to solve them. They discuss their limits, in particular the challenge of finding
a good GW mapping when there is no one-to-one correspondances in the data
(denoted as uncorrelated data splitting) and devise a benchmark to compare the
methods on words embeddings with bilingual vocabularies (which provides them
with a ground truth mapping). The benchmark show teh limits of the existing
methods whose accuracy greatly decreases with alpha the proportion of
"correlated" data. Then they propose an Neural GW solver that requires to solve
a minimax optimization problem. They show that their method outperforms the
existing ones on their benchmark especially for low "correlation"  and large
scale dataset when some competitors are failing.

**Strengths:**

+ This paper addresses an important and challenging problem of solving continuous
  Gromov-Wasserstein problems from high dimensional data.
+ The paper is well written and easy to follow, the positioning in the literature
  is clear and the proposed benchmark is well motivated.
+ The question of "correlated" vs "uncorrelated" data is very relevant in
  practice and the benchmark proposed is a good contribution to the community.
+ The proposed method is novel and shows interesting empirical results on the
  benchmark.
+ The benchmark illustrates well the limits of existing and proposed methods with
  no clear winner in all settings which is a good sign of a well designed and
  honest benchmark.
+ I really appreciated the clarity of the writing and the quality of the
  scientific steps followed in the paper. Asking a question and experimenting on
  it followed by a reasonable contribution is refreshing compared to many papers
  that just throw a new method at the wall and see if it sticks.

**Weaknesses:**

Note that these weaknesses are minor and do not impact my overall positive
opinion of the paper.

+ The choice of the word "correlation" to denote overlapping of the support of
  the true aligned samples is not well chosen since it is universally used in
  statistics for something different. Perhaps "overlap ratio" would be a better
  wording.
+ While the benchmark is interesting, it is limited to word embeddings and
  bilingual vocabularies. It would be interesting to see how the methods
  perform on other types of data and tasks, possibly simulated with known
  solution.
+ The choices of performance measures in Fig 4 could be better. All Top-k
  accuracy basically look the same whereas other "marginal quality" measures
  are provided in the appendix. It would be better to have a more diverse set of
  measures in the main text.
+ The comparison in the large scale setting to NeuralGW is a little unfair since
  other solvers rely on subsampling (minibatches) that are known to be biased wrt
  the full GW solution. Existing reference on minibatch OT show that there is a
  bias induced by minibatching that could be de-biased (see e.g. "Minibatch
  optimal transport distances; analysis and applications").
+ The proposed NeuralGW woks clearly better than existing methods in the low
  correlation setting but is is actually not as good as competitors trained on a
  much smaller sample size when the correlation is high (alpha=0.9). It is
  necessary to discuss this point a bit more in the paper.

**Questions:**

+ Could you please address the weaknesses mentioned above?

+ The Topk accuracy measures seem to nearly constant wrt K. It means that only
  the closest points are (relatively) well mapped and the others are not? Could
  you please comment on that?

---

> ### Author Response · Authors · 2025-11-22
> **Answer to Reviewer wskS [1]**
>
> We thank the reviewer for the overall review, for the good score and the encouraging comments about our research. Below we provide detailed answers to the observations and questions.
>
> **[W1] Choose of word correlation**
>
> We agree on this observation; however, unfortunately, changing this terminology would lead to major changes in the whole text as well as in several figures. On the other hand, although we agree on the fact that the term (un)correlatedness may be confused when compared to its definition in statistics. We believe that it is also suitable for our pipeline and we are confident that readers may also adopt it easily while reading our work.
>
> **[W2] While the benchmark is interesting, it is limited to word embeddings and bilingual vocabularies. It would be interesting to see how the methods perform on other types of data and tasks, possibly simulated with known solution.**
>
> We recognize the importance of evaluating our pipeline on different types of data. However, our methodology requires datasets that satisfy the following conditions: (a) the source and target distributions must be alignable, meaning they share geometric similarities; (b) the ground-truth pairs between source and target samples must be known; and (c) the dataset must be sufficiently large to train continuous models. As suggested by Reviewer hWa4, we explicitly add this information in Lines 293-295 of the revised manuscript.
>
> In most of the cases, the conditions mentioned above are not met. One example is the case of the experiment with biological data that we present in Appendix D.6. In this case, none of the GW solvers can find an optimal GW coupling as it can be seen in Table 4. This example is particularly interesting since the dataset was also used in [1] and they experimentally proved that Fused GW can find an optimal solution. This implies that not all the datasets can be aligned by the general GW solver; however, this analysis is out of the scope of our work as it considers modified versions of the GW formulation.
>
> [1] Dominik Klein, Théo Uscidda, Fabian Theis, and Marco Cuturi. Generative entropic neural optimal transport to map within and across spaces.
>
> **[W3] The choices of performance measures in Fig 4 could be better. All Top-k accuracy basically look the same whereas other "marginal quality" measures are provided in the appendix. It would be better to have a more diverse set of measures in the main text.**
>
> We thank the reviewer for this suggestion, in the revised version of our manuscript we added marginal metrics to Figures 3 and 4 instead of the additional Top-5 and Top-10 accuracies.
>
> **[W4] Minibatch training**
>
> We thank the reviewer for this suggestion. We have reviewed the proposed work [1] and its implementation. Although this method provides an interesting approach for minibatch GWOT training, it still requires computing a large cost matrix over the entire dataset. Since our dataset contains approximately $400\text{K}$ samples, this approach becomes infeasible at such scale due to memory constraints to store a cost matrix with a size of $400000\times400000\approx 10^{12}$.
>
> [1] Fatras, K., Zine, Y., Majewski, S., Flamary, R., Gribonval, R., \& Courty, N. (2021). Minibatch optimal transport distances; analysis and applications.
>
> **[W5] The proposed NeuralGW works clearly better than existing methods in the low correlation setting but is is actually not as good as competitors trained on a much smaller sample size when the correlation is high (alpha=0.9). It is necessary to discuss this point a bit more in the paper.**
>
> We thank the reviewer for the observation. As noted, the performance of our NeuralGW solver is indeed not as good as discrete methods when the correlation is high ($0.8 \le \alpha \le 1.0$). We attribute this behavior to optimization–related factors inherent to NeuralGW. More specifically, the underlying min–max–min objective is difficult to optimize in practice, and its performance remains sensitive to hyperparameter choices and initialization. In contrast, discrete GW solvers under high correlatedeness level just need to recover the correct matching.

---

> > ### Author Response · Authors · 2025-11-22
> > **Answer to Reviewer wskS [2]**
> >
> > **[Q2] The Topk accuracy measures seem to nearly constant wrt K. It means that only the closest points are (relatively) well mapped and the others are not? Could you please comment on that?**
> >
> > This behavior is entirely linked to the type of embedding used. In the case of Twitter-GloVe (Figure 4a), the gap between Top@1 and Top@10 accuracies is clearly noticeable while the use of BPEmb on the same dataset (Figure 12b) leads to a smaller difference between these same  accuracies. As the same pipeline and metrics were used in all the cases, it is possible to conclude that datasets with BPEmb represent an easier problem to solve compared to their GloVe counterparts. This conclusion is also supported by the fact that a) The obtained accuracies are close to 1 and b) baselines have good performance even for relatively low correlation levels ($0.4\le\alpha\le 0.7$).
> >
> > It is also important to point out that our revised manuscript does not include Top-5 and Top-10 accuracies for the experiments in the main text (Figures 3 and 4). We removed them considering that reporting Top-1 is informative enough and following the suggestion provided in [W3].
> >
> > **Concluding remarks.** Kindly let us know if the clarifications provided address your concerns about our work. We're happy to discuss any remaining questions during the discussion phase.

---

> > > ### Comment · Reviewer_wskS · 2025-11-27
> > >
> > > Thanks you for your responses. But I disagree rather strongly with some of them.
> > >
> > > **W1** : Your naming of "correlation" is against all standards in ML and statistics. The constant approximate renaming of things in ML is a big problem for the field and science in general and it prevent people from being able to do proper literature search. I have no doubt that a reader can adapt to any notation but you should not force it upon them. And your response to those points is basically that you won't do it because it's too much work to do a search and replace? This is not a satisfying answer.
> > >
> > > **W4** :  Doing minibatch to estimate an averaged huge OT plan in memory is the worst thing you can do and not at all what I suggested. The classical minibatch OT applications use multiple small(ish) batches to find plan and minimize the expectation of their objective wrt those batches. This means that instead of the two step procedure you use for discrete GW solvers in FlowGW or AlignGW, you would fit in average over multiple plans that in expectation cover your whole dataset with a  stochastic optimization over batches. This would make the comparison much more fair because the models would be trained on the same datasets instead of very small dataset for the numerically limited methods. This is a classical strategy used for instance for  the two reference below about generative modeling and clearly FlowGW can be trained with minibtaches using the same setup as Tong et al 2023.  And the minibatch strategy can also clearly be used instead of the two step procedure you use for AlignGW with refitting an MLP.
> > >
> > > Deshpande, I., Zhang, Z., & Schwing, A. G. (2018). Generative modeling using the sliced wasserstein distance. In Proceedings of the IEEE conference on computer vision and pattern recognition (pp. 3483-3491).
> > >
> > > Tong, A., Fatras, K., Malkin, N., Huguet, G., Zhang, Y., Rector-Brooks, J., ... & Bengio, Y. (2023). Improving and generalizing flow-based generative models with minibatch optimal transport. arXiv preprint arXiv:2302.00482.
> > >
> > > **Q2** Your response is interesting. But the fact that topk is no really sensitive to k could means that you learn to align very well part of the space but completely fail on other. It would be interesting to look at the histogram of the ranking of the true correspondence to check if this is what happens (one big dirac at rank 1 and most other ranks much larger ?) If this depends on the embedding and or method this is an interesting results that needs to be discussed.

---

> ### Author Response · Authors · 2025-12-03
> **Additional answer to Reviewer wskS**
>
> **[W1]. Usage of term "correlatedeness''**
>
> We are sorry that our answer seems to have given the wrong impression. For sure, it is not a problem for us to perform certain changes in manuscript, we were just worried about other readers which could be dissatisfied by the change of a key term used in our work. This is what we meant by "major changes''. At the same time, thanks to the additional comment by the reviewer we clearly understand the importance of the terminology point. Following the suggestion, we substitute term "correlatedeness'' with "statistical relatedness". Note that we do not use the term "overlap ratio'' proposed by the reviewer, since we found that this term also could lead to ambiguities, e.g., "overlap $= 1$'' could be wrongly treated as "source data coincides with target data''. Moreover, the term overlap seems to ignore the underlying relationships between source and target data (e.g., highly correlated pairs naturally lead to low distortion). We hope that these changes in the **new revised manuscript** address the reviewer’s concerns and improve the clarity and terminology consistency of our paper.
>
> **[W4]. Training with minibatch strategy.**
>
> It seems that there is a tiny misunderstanding. In the original manuscript, we had a method trained with minibatches as described by the reviewer (see below); therefore, when answering **W4**, we assumed we were asked about more advanced minibatch-OT techniques proposed in "Minibatch optimal transport distances; analysis and applications", which prescribe storing the full OT mapping in memory.
>
> * **FlowGW with minibatches** In fact, in our **original manuscript**, we have FlowGW trained with minibatch strategy, similar to (Tong, Fatras et. al., 2023), as described by the reviewer. Please see lines 467-469 and Figure 4 in the revised version of the manuscript. In particular, conditional flow matching objective of FlowGW was fitted on top of GW coupling between minibatches, following best practices of minibatch OT strategy.
>
> * **AlignGW with minibatches**. We would like to emphasize that AlignGW is fundamentally a discrete method, fitting an MLP on top of this method is our art to make an mvp of a continuous GW method. In principle, it is possible to fit (similar to FlowGW) a CFM on top of AlignGW. Still, we decided to stick to MLP on top of AlignGW to expand the *variety* of considered techniques: in our comparison we already have a flow-matching minibatch-OT based method (FlowGW), so it is interesting to check the MLP regression-based approach.
>
> **[Q2]. Top-k statistics.**
>
> As mentioned before, the small gap between Top@1 and Top@10 highlighted by the reviewer appears only when using BPEmb. To better understand this behavior, we conducted additional experiments. The results are now included in Table 6 and 7 from Appendix F.2 in the **new revised version** of the manuscript. In these tables we report accuracies for several values of $k$. The models trained were AlignGW (Table 6) and NeuralGW (Table 7) on Twitter-GloVE and Twitter-BPEmb for the $100\rightarrow 50$ setup. We trained five runs for each setup.
>
> **AlignGW (Table 6)**
>
> For BPEmb with higher $\alpha$, we observe near-perfect alignment already at very small $k$, so $\text{Top@k}$ increases quickly. This rapid increase does not indicate that the models fail on part of the space; rather, it reflects that all true correspondences are already captured by $k=20$. For GloVe with $\alpha = [1.0,0.5]$, accuracy improves steadily with $k$, what is a regular behavior. The only configurations that match the reviewers’ concern regarding $k$ are GloVe and BPEmb with $\alpha = 0.0$. This corresponds to a **statistically unrelated** setting, where weak performance is expected.
>
> **NeuralGW (Table 7)**
>
> For GloVe embedding, accuracies increase steadily with $k$, showing that true correspondences are spread across many ranks rather than concentrated only at Top1. The cases that partially resemble the reviewer's concern about partial alignment are BPEmb for $\alpha=[1.0,0.0]$. In these settings, $\text{Top@k}$ changes only slightly with respect to $k$.
> However, a closer look shows that this is caused by the **unstable performance** of NeuralGW in these configurations. As shown in Table 7, the standard deviation is very high in both cases. This is particularly important since for BPEmb some runs fitted the target space almost perfectly ($\text{Top@}1=1$) while others produced nearly no correct matches. This instability is what leads to the small differences observed between $\text{Top@1}$ and $\text{Top@10}$. Importantly, the case $\alpha = 0.5$, where the runs were stable, shows a much stronger sensitivity to $k$. Which additionally confirms our hypothesis regarding the connection between the small gap and stability.
>
> Overall, the observed behavior is caused by the unstable behavior of NeuralGW rather than partial matching of the target space. We hope our answer addresses the reviewer's concerns.

---

### Official Review · Reviewer_kxyJ · 2025-10-29

**Soundness:** 2
**Presentation:** 2
**Contribution:** 3
**Rating:** 4
**Confidence:** 4

**Summary:**

This paper focuses on establishing a benchmark for neural Gromov–Wasserstein (GW) problems. The authors identify key existing works addressing this task and propose a methodology for benchmarking these methods.
The main observation is that most neural GW papers operate in a correlated regime, which tends to favor their reported performance, and are (naturally) influenced by the batch size that can be processed. The proposed benchmark explicitly varies the level of correlation and evaluates the robustness of different methods under these conditions.
In addition to this benchmarking framework, the authors introduce a new neural GW approach in the inner-product setting, derived from a duality theorem. This method is designed to be more robust to correlation effects.

**Strengths:**

Overall, I find the paper well written. It is dense, but the problem is clearly formulated. The related work section is also well presented: the methods in Section 3 are described concisely yet with enough detail to understand their scope and relevance.
Identifying the correlated dataset setting is, in itself, an interesting and valuable contribution.
I find the experimental results in Section 4.2 particularly compelling, as they demonstrate that the presence of a natural pairing or correlation plays a crucial role in achieving good performance for neural GW solvers.

**Weaknesses:**

Overall impression:

Although the contributions are interesting, I find that several aspects of the paper remain unclear. The main message is somewhat blurred by a set of experiments that sometimes appear contradictory. At first, the paper suggests that the main challenge for neural GW methods lies in the level of correlation in the data; later, however, the issue seems to shift toward their inability to handle large-scale problems.
This mixture of factors within the experiments makes the overall conclusion difficult to interpret, leaving the reader somewhat frustrated by the lack of a clear and unified takeaway message.

Unclear points:

- About "Limitations of existing methods"

One of the main contributions of the paper is to show that, in the correlated regime, existing neural GW methods tend to be favored. More precisely, the correlated setting corresponds to random variables $(X, Y) \sim (i_d \times \sigma) \cdot \pi$, where $\sigma$ is a permutation and $\pi$ a coupling.

However, the train/test procedure related to this definition is not entirely clear. From what I understand, the “correlated” regime essentially corresponds to the “paired” regime — that is, a setting where there exists a natural alignment or pairing between samples.
To “break” this pairing, one can simply rematch the data points, which is indeed what the proposed procedure does. Yet, such manipulation is only feasible on synthetic datasets or on data where the natural pairing is explicitly known (e.g., word-pair datasets in embedding problems). I think it would be worth emphasizing that this procedure can only be applied in these specific cases.

- About AlignGW:

It is stated that “we train a Multi-Layer Perceptron on the barycentric mapping,” but the barycentric mapping can in principle be defined in both directions (source to target or target to source). Which direction is used here, and why was this specific choice made?

- A propos des resultats 4.2:

(a) Why were not all baselines included in Figure 3?
Including all methods would help make the comparison more comprehensive.

(b) The description of the matching metrics is somewhat confusing or hard to follow. What exactly is meant by the reference pool and reference space? A bit more formalism here would help clarify what these objects represent. Similarly, the notion of “optimal pair” could be defined more precisely.
The same applies to the marginal metrics: it seems they refer to divergences between $\mathbb{Q}$ and $T(\mathbb{P})$, but an explicit equation would make this clearer.

- About section 5.2:

The main remark here is that Figure 4 is very difficult to read. The results are not visually clear, and the curves are hard to distinguish.
Where are regGW and FlowGW in the first subfigures? Who are the “other methods”?
This is unfortunate, as this figure seems central to the paper’s conclusions, but its current presentation makes it difficult to interpret.


- About the proposed method:

It is not clear to me why the proposed method should be more robust to correlation levels. Do the authors have any intuition on this ?

- Other remarks:

The citation style is somewhat unconventional and could be standardized to match common academic formatting.
Also consider citing [1] a very recent paper on the subject.

[1] Unsupervised Learning for Optimal Transport plan prediction between unbalanced graphs, Sonia Mazelet, Rémi Flamary, Bertrand Thirion, NeurIPS, 2025.

**Questions:**

see above

---

> ### Author Response · Authors · 2025-11-22
> **Answer to reviewer kxyJ [1]**
>
> Dear reviewer. Thank you for your review. We carefully address your questions and observations below.
>
> **[General Weakness:] Although the contributions are interesting, I find that several aspects of the paper remain unclear. The main message is somewhat blurred by a set of experiments that sometimes appear contradictory.**
>
> We appreciate the reviewer’s comment. As correctly noted, our central message is to highlight the importance of incorporating the level of correlatedness when evaluating GW solvers. In order to study how this factor affects different solvers, we consider both discrete and continuous approaches.
>
> Discrete solvers typically produce solutions that depend on the amount of training data. Although the discrete methods we evaluate can perform out-of-sample inference and show good performance on unseen data (Figure 3), this can be misleading. Their success relies on training with paired or correlated data, an assumption that is rarely realistic in practical applications where ground-truth correspondences are not available.
>
> Therefore, we tested the discrete solvers on uncorrelated data and observed a significant drop in performance. While they still produce a coupling, it is far from optimal. We attribute this to the limited amount of training data available to discrete solvers, which prevents them from capturing the underlying geometric structure of the source and target distributions, particularly in higher dimensions. In contrast, NeuralGW can recover the correct structure when trained on "large-scale" data, as discussed in Appendix F.1, where we further explain the geometric reasons behind this behavior. Therefore, the ability to train on "large-scale" data is the key component to beat the "curse of correlatedeness".
>
> As a conclusion, the failure of discrete solvers on more realistic (uncorrelated) setups comes from their inability to handle larger datasets. Even though discrete solvers are good for certain application, we highlight the need for a continuous solver that can properly deal with uncorrelated data.
>
> **[W1] About "Limitations of existing methods"**
>
> We appreciate the reviewer's comment as it precisely describes our benchmark process. Indeed, it is true that our proposed benchmark construction can only be applied on certain datasets where: 1) ground-truth pairs are known, 2) source and target datasets are alignable and 3) they are large enough to train continuous models. Given such considerations, we can only consider a small number of datasets for our benchmark. We make this clearer in the revised version of the manuscript by emphasizing these three aspects in Section 4.1.
>
> In spite of this, we expect that our conclusions may be extended to setups with alignable structures (e.g., single-cells [1, 2], graph completion [3]) even if such datasets are not suitable for our benchmark due to unknown ground-truth pairs or insufficient size.
>
> [1] Demetci, P., Santorella, R., Sandstede, B., Noble, W. S., \& Singh, R. (2020). Gromov-Wasserstein optimal transport to align single-cell multi-omics data.
>
> [2] Dominik Klein, Théo Uscidda, Fabian Theis, and Marco Cuturi. Generative entropic neural optimal transport to map within and across spaces.
>
> [3] Xu, H., Luo, D., Zha, H., \& Carin, L. (2019). Gromov-Wasserstein learning for graph matching and node embedding. 36th International Conference on Machine Learning, ICML 2019.
>
> **[W2] About AlignGW**
>
> The initial choice of barycentric projection was done on the target, i.e. $\mathbf{y_{pred}}=(1/\sum_i T_{ij})\sum_{j=1}^{N{y}}T_{ij}y_j$, as our metrics are computed on this prediction.
>
> Nevertheless, in the latest version of our paper we replaced the barycentric projection for an argmax strategy: First we compute ```idx=np.argmax(T, axis=1)``` and then the chosen target samples are ```y\_pred=y[idx]```. We made this change as it gave us better performance for the method. In the revised version of our manuscript we mention this peculiarity in the description of AlignGW and in Appendix E with more details.
>
> **[W3-Q1] Why were not all baselines included in Figure 3? Including all methods would help make the comparison more comprehensive.**
>
> We did not include results for RegGW and CycleGW in Section 4.2 since experiments in this section consider datasets with $3\text{K}$ samples what is not enough to train continuous models. In order to show this, we trained these models for five runs on the totally correlated setup ($\alpha=1$) and the results can be found in the table below
>
> | Method  | Top@10        | FOSCTTM      | Distortion    |
> |---------|----------------|---------------|-----------------|
> | CycleGW | 0.002$\pm$1e-5 | 0.45$\pm$0.02 | 0.031$\pm$0.001 |
> | RegGW   | 0.002$\pm$1e-5 | 0.45$\pm$0.01 | 0.057$\pm$0.002 |
>
> We explicitly report this poor performance in lines 294–296 of the original version, and we believe that including these results in the plots would reduce their clarity.

---

> ### Author Response · Authors · 2025-11-22
> **Answer to reviewer kxyJ [2]**
>
> **[W3-Q2] The description of the matching metrics is somewhat confusing or hard to follow. What exactly is meant by the reference pool and reference space? A bit more formalism here would help clarify what these objects represent. Similarly, the notion of “optimal pair” could be defined more precisely. The same applies to the marginal metrics: it seems they refer to divergences between $\mathbb{Q}$ and $T(\mathbb{P})$, but an explicit equation would make this clearer.**
>
> We believe the reviewer refers to Lines 913-917 in Appendix C.1 from the original manuscript. In this case reference pool and reference space are the same object. We understand the confusion and the revised version uses the same name in the main text and in the Appendix.
>
> On the other hand, the **optimal pairs** are the ground truth pairs. A ground truth pair is known by construction as the source and target distributions contain the same words, but the dimension of embedding is different. In the table below we verify that these ground-truth pairs are also optimal pairs. In particular, we compare the *distortion* $\int | c_x(x, x') - c_y(T(x), T(x')) |^2 dP(x) dP(x')$ for $T$ given by GT pairing of word embeddings and those obtained by fitting discrete GW solver from POT library on the same data.
>
>
> | Dataset| $\alpha$ | Distortion-GT (mean) | Distortion-GT (std) | Distortion-DGW (mean) | Distortion-DGW (std) |
> |-----------------------------|-------|------------------------|----------------------|------------------------|------------------------|
> | Twitter-Glove (100$\rightarrow$50)| 1.0 | 0.014068719| 8.49e-05 | 0.014215  | 0.000320 |
> | MUSE-BP (100$\rightarrow$50)| 1.0 | 0.010357639| 7.20e-05 | 0.01037014| 4.49e-05 |
>
> The metrics are computed by averaging distortions over batches of the 400k ground-truth pairs.
>
> The reviewer is correct in the case of **marginal metrics**. All of them (Sinkhorn divergence, MMD and BW-UVP) are computed between $\mathbb{Q}$ and $T_{\sharp}\mathbb{P}$. In Appendix C of the revised version we provide explicit equations for matching and marginal metrics. We hope this change helps to clarify this point.
>
> **[W4] ... Figure 4 is very difficult to read. The results are not visually clear, and the curves are hard to distinguish. Where are regGW and FlowGW in the first subfigures? Who are the “other methods”? ...**
>
> We thank the reviewer for the observations as they will help to improve the clarity of our results.
>
> - **Revised Figures.** The revised version of our manuscript includes plots that more clearly distinguish the performance of the solvers. More precisely RegGW, CycleGW and FlowGW\_m use different line styles in Figure4. We also have to mention that the plots now include some of the marginal metrics as requested by Reviewer wskS.
>
> - **Regarding the "Other methods" legend.** In Line 430, we state that by "Other methods" we refer to the baseline solvers that we tested in Section 4.2. Nevertheless, we understand that this legend may be misleading, therefore, we update our plots and text to use the legend "Baseline solvers" instead.
>
> **[Q1] Why NeuralGW is robust with respect to correlation**
>
> The robustness of NeuralGW with respect to the correlatedness $\alpha$ is inherited from the formulation of our solver, which aims to minimize equation 6. Let's represent $X_b$ and $Y_b$ as source and target batches, respectively. Therefore, the only place where the interconnections between $X_b$ and $Y_b$ may influence the overall optimisation is the intermediate maximization w.r.t. potential $f$: $\frac{1}{|X_b|}\sum_{x_b \in X_b} f(T(x_b)) - \frac{1}{|Y_b|} \sum_{y_b \in Y_b} f(y_b) \rightarrow \max_{f}$. Due to the average over batch and the application of the potential $f$, the possible (un)correlatedness of $X_b$ and $Y_b$ has almost no effect on the final performance.
>
> Below, we also provide comments regarding the behavior of other continuous solvers with respect to the correlatedness levels.
>
> - **RegGW:** During the training, the method computes the so-called Gromov-Monge gap regularizer, which directly depends on the solution to \underline{ the discrete GW solution between batches $X_b$ and $Y_b$}. This explains why the performance of RegGW is affected by the correlatedness between $X_b$ and $Y_b$.
>
> - **FlowGW\_mb** is also built on top of GW solution between minibatches $X_b$ and $Y_b$, explaining the behavior of the performance.
>
> - **CycleGW.** The method's objective consists of several terms, including inter-space discrepancy $\Delta_{x, y}(f, g) = \vert c_{\mathcal{X}}(X_b, g(Y_b)) - c_{\mathcal{Y}}(f(X_b), Y_b)|$ (averaged over batches), see CycleGW paper [77], Subsection 3.1. Here, $f$ and $g$ are learned forward (and inverse) GW mappings between the spaces $\mathcal{X}, \mathcal{Y}$. This term is clearly influenced by the correlation between $X_b$ and $Y_b$. We conjecture this is the reason for the observed behavior of the performance.

---

> > ### Author Response · Authors · 2025-11-22
> > **Answer to reviewer kxyJ [3]**
> >
> > **[W5] Issue with citations.**
> >
> > We apologize for this issue, it was a problem that we did not notice until the submission deadline passed. This is fixed in the revised version. We also properly cite the suggested paper (Mazelet, 2025) in the Introduction in Line 65 of the revised version. Thanks for pointing it out.
> >
> > **Concluding remarks.** Kindly let us know if the clarifications provided address your concerns about our work. We're happy to discuss any remaining questions during the discussion phase. If you find our responses satisfactory, we would appreciate it if you could consider raising your score.

---

### Official Review · Reviewer_vEr5 · 2025-10-29

**Soundness:** 2
**Presentation:** 2
**Contribution:** 2
**Rating:** 4
**Confidence:** 3

**Summary:**

This paper has a twofold purpose: to provide a benchmark for Gromov–Wasserstein (GW) solvers, and to propose a new continuous Monge–GW method that does not rely on discrete techniques. Regarding the first goal, the authors report that existing solvers perform well when correlated data is available but fail in the absence of correlation. To address this limitation, they introduce a new methodology (NeuralGW), which is designed to handle fully uncorrelated setups and is claimed to be practically useful in more realistic and challenging scenarios. However, this new solver requires large amounts of training data, and other remaining challenges leave the door open for further research.

**Strengths:**

This paper highlights the limitations of existing GW solvers: in the interplay between discrete and continuous approaches, current methods are said to rely more heavily on the empirical (discrete) side rather than the continuous perspective, which, in practice, should treat data as i.i.d. samples.

**Weaknesses:**

- In general, the presentation is not clear, making it difficult to fully understand the core problem.

- Five existing GW methods are listed in Section 3, but only three of them are analyzed in Section 4.2.

- Although the authors reference the seminal paper by Dumont et al. on the existence of Monge-GW assignments, the distinction between finding an optimal coupling/plan and finding an optimal map (i.e., the Kantorovich vs. Monge formulations) is blurred.

- The quality of Figure 2 is poor; it appears blurry.

- The beginning of Section 4.1 (lines 222–246) is unclear.

- The proposed solver appears to have several drawbacks.

**Questions:**

- I suggest formalizing what is meant by a “continuous setup.”

- In the classical OT problem, the Kantorovich formulation turns the optimization into a linear program, making it more tractable than the Monge formulation. Could the authors elaborate on how this comparison translates to the GW setting?

**Details Of Ethics Concerns:**

No concerns.

---

> ### Author Response · Authors · 2025-11-22
> **Answer to Reviewer vEr5 [1]**
>
> Dear reviewer, thank you for spending time reviewing our paper. Below, we provide answers and insights addressing your observations and questions.
>
> **[W1] In general, the presentation is not clear, making it difficult to fully understand the core problem.**
>
> In general words, in our paper we aim to call the attention of the community towards a gap we found on current evaluation pipelines for GW solvers. More specifically, current evaluation setups do not consider the level of correlation/pairness between the source and the target distributions. This is particularly important in the case of GW solvers as they aim to match the geometry of the distributions rather than solving a matching problem between samples. To address this, we propose a benchmark that explicitly incorporates varying levels of (un)correlatedness between distributions and evaluate several solvers to assess how significantly this factor affects their performance.
>
> We understand that the paper may be challenging to follow, as this is not a standard topic. We hope our response clarifies the general goal of the work, and if not, we would appreciate it if the reviewer could indicate which specific aspects remain unclear.
>
> **[W2] Five existing GW methods are listed in Section 3, but only three of them are analyzed in Section 4.2.**
>
> We did not include results for RegGW and CycleGW in Section 4.2 since experiments in this section consider datasets with $3\text{K}$ samples what is not enough to train continuous models. In order to show this, we trained these models for five runs on the totally correlated setup ($\alpha=1$) and the results can be found in the table below
>
> | Method  | Top@10         | FOSCTTM       | Distortion      |
> |---------|----------------|---------------|-----------------|
> | CycleGW | 0.002$\pm$1e-5 | 0.45$\pm$0.02 | 0.031$\pm$0.001 |
> | RegGW   | 0.002$\pm$1e-5 | 0.45$\pm$0.01 | 0.057$\pm$0.002 |
>
> We explicitly reported this poor performance in lines 294–296 of the original version, and we believe that including these results in the plots would reduce their clarity.
>
> **[W3] Distinction between Monge and Kantorovich; [Q2] "In the classical OT problem, the Kantorovich formulation turns the optimization into a linear program, making it more tractable than the Monge formulation. Could the authors elaborate on how this comparison translates to the GW setting?"**
>
> We thank the reviewer for this point and question, it seems there is a tiny misunderstanding, which requires certain clarifications. The answer/clarification depends on the **data setup** under which we consider our OT problem (see Subsection 2.3 of our paper).
>
> **Discrete setup**. In this case, as noted by the reviewer, people indeed typically consider Kantorovich formulation, not Monge version, to have linear problem and to guarantee the existence of the solution of the classical OT. In case of GWOT (eq. 2 in our paper), the situation is similar: "Kantorovich's'' version is a quadratic program (QP), and the existence of "Monge's'' solution is established only under certain assumptions plus **absolute continuity** of the source/target measures, see [1]. To our knowledge, there are no practical works which consider discrete Monge's GW.
>
> **Continuous setup**. Situation is different. No more linear/non-linear, QP/non-QP programs (these terms are not applicable under this setup), the existence under certain assumptions holds true both for Kantorovich's and Monge's version. The practical difference is also minimal - one can switch between versions by *noise outsourcing* (see [2]), i.e., by additional noisy input of the trained mapping, which allows learning conditional distributions, not deterministic one-to-one mappings. We believe that this machinery is unnecessary for our NeuralGW model. From a theoretical perspective, the ground-truth GW solution is guaranteed to be a deterministic mapping. We expect that the learned model will ignore additional noise inputs and not benefit from them.
>
> [1] Dumont et. al., (2025). "On the existence of Monge maps for the Gromov–Wasserstein problem." FCM
>
> [2] Koroting et. al., (2023), "Neural optimal Transport." ICLR
>
> We kindly ask the reviewer if our clarifications address their question, and ready for follow-up discussions, if needed.

---

> ### Author Response · Authors · 2025-11-22
> **Answer to Reviewer vEr5 [2]**
>
> **[W4] The quality of Figure 2 is poor; it appears blurry.**
>
> The revised text includes a higher resolution image for this Figure.
>
> **[W5] The beginning of Section 4.1 (lines 222–246) is unclear.**
>
> In the first paragraph of Section 4.1, we introduce the notions of correlated and uncorrelated setups and describe what we expect the solvers to learn in each case. A correlated setup is created by forming subsets of source-target pairs with one-to-one correspondences and then randomly shuffling the target data. Although training remains unpaired, the dataset still contains a shuffled version of the correct optimal pairs (those with minimal distortion). In contrast, an uncorrelated setup uses no ground-truth pairs at all; training samples are drawn independently from the underlying source and target distributions.
>
> Ideally, in both scenarios, GW solvers are expected to recover correct GW mapping between original source and target distributions (which corresponds to ground truth pairing and yields minimal distortion). However, our extensive experimental study shows that it is not always the case. In particular, discrete solvers on small amount of data may recover a GW coupling, but not between the *original* distributions but rather between *sample* distributions.
>
> We hope our answer clarifies this point. If not, we kindly ask the reviewer to be more specific about what part of the paragraph remains unclear.
>
> **[W6] The proposed solver appears to have several drawbacks.**
>
> Although our proposed NeuralGW solver exhibits instability, it is promising because it is the only method whose performance does not depend on the level of correlatedness. This, in turn, highlights the true potential of a fully continuous GW solver.
>
> **[Q1] I suggest formalizing what is meant by a “continuous setup.”**
>
> We kindly refer the reviewer to Subsection 2.3 where we formally introduce the continuous setup. In summary, under discrete setups we solve GW between discrete sample distributions supported on available samples (suitable for small datasets), and aim to recover a *matching* (correspondence) between them. Continuous setups, on the other hand, solve GW between underlying distributions by using random batches from them. The key here is that we aim to recover a (generalizable) *mapping* between the distributions, not a matching between certain samples.
>
> To improve clarity, the revised version replaces the word “formulation” with “setup,” since we agree that the original wording may have caused confusion.
>
> **Concluding remarks.** Kindly let us know if the clarifications provided address your concerns about our work. We're happy to discuss any remaining questions during the discussion phase. If you find our responses satisfactory, we would appreciate it if you could consider raising your score.

---

### Official Review · Reviewer_hWa4 · 2025-11-02

**Soundness:** 2
**Presentation:** 3
**Contribution:** 2
**Rating:** 2
**Confidence:** 3

**Summary:**

The Gromov-Wasserstein distance is a metric commonly used in practice to compare two different metric measure spaces due to its nice geometric interpretation of computing the most isometric map between the spaces. While nice for its mathematical properties, the Gromov-Wasserstein distance is quite challenging to compute. Existing methods for computing the Gromov-Wasserstein distance assume discrete metric measure spaces, and therefore fail to capture the underlying maps between continuous distributions that samples come from. To extend the Gromov-Wasserstein distance to the continuous setting, algorithms then rely on a correlated sample between the two distributions.

The authors of this work experimentally suggest that existing methods for computing Gromov-Wasserstein distances between correlated samples, i.e. samples where the optimal pairs have some underlying association in the metric measure spaces, fail to extend to the continuous Gromov-Wasserstein distance when discrete samples are taken i.i.d. from the two distributions. They then design a neural network which computes continuous Gromov-Wasserstein distances without relying on discrete instances like in prior works and empirically verify that their neural network outperforms existing methods for computing continuous Gromov-Wasserstein distances.

**Strengths:**

The introduction to the Gromov-Wasserstein problem was pretty well written and easy to follow.

The overview of existing algorithms for continuous Gromov-Wasserstein distance in section 3 was very useful and also well written.

 The authors clearly describe a problem with existing algorithms for the Gromov-Wasserstein distance and design a neural network which outperforms existing algorithms.

**Weaknesses:**

The conclusion "existing algorithms therefore don't work for uncorrelated data" from Section 4 cannot be made as a general statement. In particular, their experiments were conducted on only two data sets with very similar types of data (word to vector embeddings) and no proof is provided to justify the strong claims they make about existing work. To my understanding, provable guarantees of existing algorithms for Gromov-Wasserstein distance are lacking, i.e. existing algorithms serve as merely heuristics already. Why should one expect that the results  from these two data sets to extend to other data sets on Euclidean spaces? Why could it not be an issue with the structure of the word embedding data tested?

The authors note on line 464 that the performance of their neural network is "inconsistent with respect to initialization parameters" and "sees high standard deviation among repetitions".

**Questions:**

None

---

> ### Author Response · Authors · 2025-11-22
> **Answer to Reviewer hWa4 [1]**
>
> We thank the reviewer for the review and comments. We will address the weaknesses and questions below.
>
> **[W1] The conclusion "existing algorithms therefore don't work for uncorrelated data" from Section 4 cannot be made as a general statement. In particular, their experiments were conducted on only two data sets with very similar types of data (word to vector embeddings) and no proof is provided to justify the strong claims they make about existing work.**
>
> We believe the reviewer’s conclusion: “existing algorithms don’t work for uncorrelated data” is not accurate as it does not reflect our main message. As clarified in our Disclaimer (lines 343–347 in the original version), all discrete solvers can recover a GW coupling when used correctly, even in uncorrelated setups. The issue is that they must be trained on a small subset of samples, which may not sufficiently represent the full geometry, leading to poor generalization. Further details are explained in Appendix F.1.
>
> A simple way to avoid this limitation is to use methods that can deal with large amount of data to catch the geometry of the source/target distributions. These are continuous solvers. However, the majority of current continuous solvers still rely on discrete solvers during training and thus inherit the same limitation. To conclude, discrete solvers remain suitable for certain applications, but our results highlight the need for a fully continuous GW solver capable of handling uncorrelated settings, similar to our NeuralGW approach.
>
> **[W2] To my understanding, provable guarantees of existing algorithms for Gromov-Wasserstein distance are lacking, i.e. existing algorithms serve as merely heuristics already.**
>
> We thank the reviewer for raising this point. As noted earlier, our aim is to highlight gaps in current GW experimental setups rather than questioning the effectiveness of existing solvers. Accordingly, providing guarantees for these methods falls outside our scope. This is also problematic since each approach operates under different formulations of the GW problem.  Moreover, in most of the cases, their theoretical guarantees have already been established and extensively studied in their respective original works.
>
> **[Q1] Why should one expect that the results from these two data sets to extend to other data sets on Euclidean spaces?**
>
> In order to answer this question is relevant to clarify the following points:
>
> - We work on datasets that have **alignable structures**, this means that the source and target distributions share structural similarities that are suitable for alignment. We work under the assumption that this structure makes possible to find an optimal coupling even when the solvers are provided with data that is not directly paired or correlated. This assumption is verified by our experiments, more specifically, our NeuralGW solver can successfully model a general transport plan when trained on uncorrelated batches.
>
> - We understand that considering more setups is important; however, only a few datasets are suitable for our benchmark. More specifically, we require datasets with the following properties: a) Source and Target with alignable structures, b) known ground-truth pairs for computing the metrics and c) large enough to train the continuous solvers. We add this information in Lines 293-295 of the revised version.
>
> In summary, as our assumptions coincide with our experimental results, we expect that this behavior may hold for other type of datasets that poses alignable structures. Even though we only consider word embeddings, we believe that our findings are relevant for the community as it properly highlights key limitations of existing evaluation pipelines.

---

> ### Author Response · Authors · 2025-11-22
> **Answer to Reviewer hWa4 [2]**
>
> **[Q2] Why could it not be an issue with the structure of the word embedding data tested?**
>
> We can ensure that the problem is not in the structure of the data tested due to the following reasons:
>
> - **NeuralGW performs consistently across all setups.** Our experimental results show that NeuralGW achieves high accuracy and low distortion in every scenario, including uncorrelated cases. This holds across different embedding types and dimensionalities, indicating that the data does not introduce difficulties that prevent the recovery of an optimal coupling.
>
> - **The ground-truth pairs in the data are also optimal**. The Twitter and MUSE datasets allow to construct datasets with one-to-one correspondences as the same words can be embedded on different dimensions. These correspondences are what we refer to as “ground-truth” pairs. In the table below we verify that these ground-truth pairs are also optimal pairs. In particular, we compare the *distortion* $\int | c_x(x, x') - c_y(T(x), T(x')) |^2 dP(x) dP(x')$ for $T$ given by GT pairing of word embeddings and those obtained by fitting discrete GW solver from POT library on the same data.
>
> | Dataset                     | $\alpha$ | Distortion-GT (mean) | Distortion-GT (std) | Distortion-DGW (mean) | Distortion-DGW (std) |
> |-----------------------------|-------|------------------------|----------------------|------------------------|------------------------|
> | Twitter-Glove (100$\rightarrow$50)      | 1.0   | 0.014068719            | 8.49e-05             | 0.014215               | 0.000320               |
> | MUSE-BP (100$\rightarrow$50)            | 1.0   | 0.010357639            | 7.20e-05             | 0.01037014             | 4.49e-05               |
>
> The metrics are computed by averaging distortions over batches of the 400k ground-truth pairs.
>
> Thus, the consistent success of NeuralGW in uncorrelated setups, combined with the fact that the datasets admit optimal ground-truth correspondences, confirms that the data itself is not the source of the "issue" regarding GW solvers' performance. The low distortion of these ground-truth pairs demonstrates that the underlying geometry is alignable, and therefore the datasets do not present any structural limitations that would prevent the recovery of a correct GW coupling.
>
> **[W3] NeuralGW performance**
>
> We understand the comment by the reviewer, as we mentioned in the paper, the performance of our NeuralGW solver depends on the initialization parameters. This is unfortunately not easily fixable due to the min-max-min optimization. Nevertheless, it is important to point out that NeuralGW is the only solver that is able to learn an optimal coupling on uncorrelated data. Therefore, we believe that including results for our solver is important as it shows that our benchmark is indeed solvable and that continuous GW solvers are a promising alternative.
>
> **Concluding remarks.** Kindly let us know if the clarifications provided address your concerns about our work. We're happy to discuss any remaining questions during the discussion phase. If you find our responses satisfactory, we would appreciate it if you could consider raising your score.

---

> ### Comment · Reviewer_hWa4 · 2025-11-25
> **Acknowledging  authors' response**
>
> We thank the authors for their thoughtful response. My rating of the paper remains the same.

---

> > ### Author Response · Authors · 2025-11-25
> > **Answer to acknowledgment**
> >
> > We thank the reviewer for acknowledging our response. We believe that we have addressed the majority of the concerns and questions raised. We respectfully invite the reviewer to continue the discussion by providing any further questions or clarifications that would help us identify any remaining issues in the manuscript.

---

### Author Response · Authors · 2025-12-03
**General Comments**

We thank all reviewers for their constructive feedback throughout the review process. The main concerns raised during the rebuttal stage were centered around the clarity of our work. In response, we provide a **new revised** manuscript with the following changes:

- **Terminology update (“correlatedness”).** Following Reviewer wskS’s suggestion, we replaced the term “correlatedness’’ with “statistical relatedness’’ to avoid confusion with its standard meaning in statistics and to improve conceptual clarity.

- **Additional appendix.** Reviewer wskS also raised questions about the small gap between the Top@1 and Top@10 accuracies in the Twitter-BPEmb setup. We clarified this point in the discussion below and added Appendix F.3 to the **new revised** manuscript, as we believe this is an important point to address.

- **Improved plots and figures.** As suggested by Reviewers vEr5 and kxyJ we improved the resolution of Figure 2 and improved the clarity of our plots by choosing more suitable line types in Figure 4.

- **Improved metrics.** As suggested by Reviewer kxyJ, we provide more formal and detailed explanations about our metrics in Appendix C.1.

Altogether, we believe these updates strengthen the presentation and make the contributions of the paper easier to follow.

---

### Meta-Review · Area_Chair_5cVf · 2026-01-03

**Summary:**

The paper presents a benchmark for evaluating continuous Gromov–Wasserstein solvers under varying levels of statistical relatedness between source and target distributions, and introduces NeuralGW, a fully continuous solver that avoids discrete-in-the-loop training and appears empirically more robust when training data are fully unpaired. While the benchmarking perspective is timely and the stress-test framing is potentially useful, several major weaknesses remain, even after a constructive rebuttal. First, the empirical scope is limited, with most conclusions drawn from a narrow set of datasets (notably word embeddings), and although the authors clarified in rebuttal that they do not claim categorical failure of discrete solvers, this clarification mainly narrows the claims rather than demonstrating broader external validity across domains. Second, NeuralGW itself suffers from substantial optimization instability and high variance across initializations; the rebuttal acknowledges this and provides a rationale for why the method can still succeed in extreme unpaired regimes, but this explains the limitation rather than resolving it, leaving robustness a serious concern. Third, the conceptual framing and presentation were initially unclear, particularly regarding the distinction between Monge and Kantorovich formulations, the meaning of “continuous” training, and the role of statistical relatedness versus data scale; while the authors’ rebuttal addresses many of these points through clarified definitions, revised terminology, and more explicit metric formulations, the overall narrative still depends heavily on careful rewriting to avoid conflating problem structure with training-budget constraints. Finally, although baseline coverage, evaluation protocols, and diagnostics were improved (with added tables, clearer metrics, and additional analyses), some fairness and interpretability issues remain, such as uneven baseline treatment and reliance on coarse accuracy metrics where richer diagnostics were suggested. In summary, the authors made a genuine effort in rebuttal to clarify scope, terminology, and evaluation details, and several concerns were partially addressed, but the combination of limited empirical breadth, instability of the proposed solver, and reliance on heuristic explanations means that the paper does not yet meet the bar for acceptance in its current form.

**Reviewer Concerns:**

The rebuttal meaningfully addressed several presentation and protocol issues raised by reviewers (clearer definition of the “continuous setup”, improved figure readability, more explicit metric descriptions, added ablations/tables, and a terminology fix from “correlatedness” toward “statistical relatedness”). It also partially addressed conceptual questions about Monge vs. Kantorovich by clarifying what is meant under discrete vs. continuous setups, and it corrected some baseline reporting/legend issues. However, key concerns remain outstanding: (i) external validity (most evidence remains centered on word-embedding benchmarks with constrained applicability), (ii) NeuralGW’s optimization instability and sensitivity to initialization (acknowledged but not mitigated), and (iii) lingering fairness/interpretability questions around baseline comparability and whether coarse Top-k style metrics obscure failure modes (even though additional diagnostics were promised/added in appendices).

**Reviewer Scores:**

- [hWa4] (initial score: 2): No change (the reviewer explicitly maintained the same rating after the authors’ response).
- [vEr5] (initial score: 4): Likely no change; clarifications and figure improvements address several criticisms, but the reviewer’s core reservations about clarity and conceptual blurring would probably persist without stronger re-framing and broader evidence.
- [kxyJ] (initial score: 4): Likely no change; the rebuttal responds directly to metric formalism, missing baselines, figure legibility, and citations, but the reviewer’s broader concern about a mixed/blurred takeaway (correlation vs. scale vs. methodology) would plausibly remain.
- [wskS] (initial score: 8): Slight decrease or no change; the terminology issue was ultimately corrected and the minibatch-training misunderstanding was addressed, but the reviewer’s later comments indicate strong dissatisfaction with parts of the rebuttal and a desire for deeper diagnostics, so their enthusiasm would likely be tempered even if still broadly positive.

---

### Decision · Program_Chairs · 2026-01-26

Reject